# CUPID in the Model Zoo: Online Matchmaking for Selecting Your Dream LLM

**Son Nguyen** [1]   **Xinyuan Liu** [1]   **Ransalu Senanayake** [1]

## Abstract

Users increasingly face the challenge of selecting an appropriate LLM for a given task from a rapidly growing pool of LLMs, each with distinct but often opaque latent properties. Compounding this challenge, users may lack the vocabulary or awareness to explicitly articulate the characteristics they value in an LLM's responses or deployment. We propose an interaction-efficient active learning framework in which a dueling bandit algorithm iteratively selects pairs of LLMs, collects user feedback about their responses, and updates its belief about the user's latent preferences. We introduce a novel belief-aware upper confidence bound strategy that balances exploration of the model pool with exploitation of inferred preferences, enabling efficient alignment between user needs and LLM capabilities under user-specified cost and time budgets. Through diverse experiments on LLMs and human studies, we experimentally verify that our model can efficiently match well-aligned LLMs to users at a lower cost.

## 1. Introduction

Large language models (LLMs) are rapidly proliferating into enterprise workflows and everyday user needs, where different applications demand distinct LLM capabilities. For example, an LLM supporting a small clinical practice must balance medical competence with administrative reasoning, financial literacy, and the ability to interact with insurance and billing tools as an agentic AI. As the number of LLM variants and providers continues to grow, selecting an appropriate LLM has become increasingly difficult for end users, raising challenges around trust, reliability, and deployment cost. Beyond raw accuracy, users must now consider multiple objectives—including domain specialization, cost,

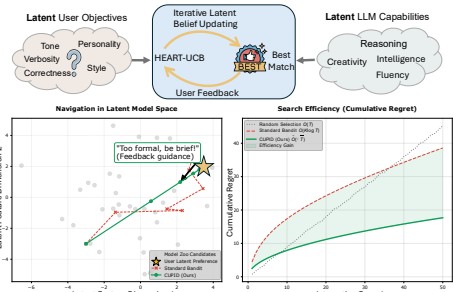

*Figure 1.* To help users with *latent objectives* to find matching LLMs with *matching capabilities*, CUPID efficiently navigates a latent preference space. This targeted navigation results in lower cumulative regret, allowing the system to quickly identify the optimal model at a lower cost.

latency, safety, and response characteristics—which are often difficult to formalize upfront as a traditional software requirements specification (SRS). Moreover, unlike traditional machine learning models, LLMs function as dynamic, responsive, intelligent agents, and hence, users have to engage with them directly to determine whether they truly meet their needs.

We argue that an effective LLM matching system in this setting should satisfy the following desiderata:
 **Desideratum 1 (Application-centric)**: The system should interact with an application-specific user (or team) to infer individual preferences, rather than relying on generic preferences aggregated from large user populations.
 **Desideratum 2 (User-centric)**: The system should leverage cognitively lightweight forms of user feedback (e.g., clinicians and admins) to support expressive model comparisons over multiple, but minimal, interaction turns.
 **Desideratum 3 (Budget-aware)**: The system should explicitly respect user-specified time and cost budgets available for the LLM matching process. This requirement is expected to become more pronounced in expensive agentic AI systems with long horizon planning and multiple tools as well as long video generation and world models.

Current matching solutions primarily focus on query-level routing, in which individual prompts are dynamically assigned to models at inference time based on various tradeoffs, a setting commonly referred to as LLM routing (Chen et al., 2024; Ding et al., 2024; Chiang et al., 2025). This line

---

[1]School of Computing and Augmented Intelligence, Arizona State University, Tempe, United States of America.. Correspondence to: Son Nguyen <snguye88@asu.edu>.

*Proceedings of the 43rd International Conference on Machine Learning*, Seoul, South Korea. PMLR 306, 2026. Copyright 2026 by the author(s).

of work has demonstrated practical impact in large-scale deployments, where providers must optimize per-query efficiency under tight latency and cost constraints. However, routing methods typically operate at the level of isolated queries and assume a fixed or LLM provider-defined notion of utility. Further, routing assumes that utility is either fixed or externally specified (e.g., accuracy, latency) and optimizes each query independently. As a result, routing systems do not model persistent, user- or application-specific latent preferences. In contrast, our goal is to identify one LLM for sustained deployment for an application, using limited interactive comparisons under budgets (e.g., for deployments such as in clinical settings where one must select a compliant LLM for sustained use). Our work targets this complementary regime by focusing on user- and application-level LLM matching, rather than per-query routing, enabling efficient discovery of LLMs that best satisfy individual user needs and preferences (Hutagalung et al., 2025).

As depicted in Figure 2, we introduce *a framework*, Cost-aware User-centric Preference Identification and Discovery (**CUPID**). Since the user's objectives when making a decision are unobserved, we model objectives as a probabilistic belief over a latent variable, which is iteratively updated via a derived update rule. To ensure that the users explore different LLMs sufficiently while prioritizing those that appear most promising, we maximize an upper confidence bound (UCB) over a pool of LLMs. Concretely, CUPID maintains a Gaussian process over latent preference utilities and optimizes a novel budget-aware UCB, derived in this work, using both pairwise preference feedback and optional natural language feedback on LLM responses. This design satisfies Desideratum 1 by learning preferences at the level of an individual user or application rather than relying on population-level aggregates, Desideratum 2 by requiring only lightweight pairwise comparisons and optional language feedback, and Desideratum 3 by explicitly incorporating cost and time constraints into model selection.

To the best of our knowledge, this is the first attempt to (i) formulate latent dueling bandits using UCB and (ii) integrate language feedback into online interaction for LLM-user matchmaking. Our main contributions are as follows:

1. We formulate *CUPID* as a general *framework* for efficiently matching the user to their favored LLM for a particular application, while respecting the user's budget and time availability for the matchmaking process.

2. To do this, we derive *HEART-UCB*, a theoretically grounded online UCB *algorithm* that accounts for latent objectives and language feedback.

3. We demonstrate the effectiveness of CUPID through extensive *experiments* across baselines and human studies on language and vision modalities.

## 2. Related Work

**The problem.** A range of approaches have been proposed to route individual queries through low-cost LLMs. FrugalGPT (Chen et al., 2024) leverages an LLM router, an answer scorer, and a stop judger to learn a cascading strategy that only calls strong models when needed. Hybrid LLM (Ding et al., 2024) uses an encoder model to route easy or hard queries. Other works rely on preference (Ong et al., 2025) or observational data (Tsiourvas et al., 2025) to train routing policies that map queries to the most suitable models. Some approaches leverage ensembles (Jiang et al., 2023) or recommender systems that have access to model specifications (Zhang et al., 2025). These methods differ from ours in three key ways. First, they assume that model capabilities are known or can be reliably estimated from population-level data, whereas we treat them as latent and reveal them only through interaction. Second, they focus on *online query-to-LLM routing*, a fundamentally different paradigm from our user-to-LLM matching problem over sustained interaction. Third, routing optimizes per-query cost–quality trade-offs, whereas our goal is to identify the best-matching LLM under explicit cost and interaction budgets, minimizing user effort during the matching process.

**The solution.** Multi-armed bandits (MAB) (Robbins, 1952) refer to sequential decision problems where a decision-maker chooses among multiple arms with unknown rewards. Yue et al. (2012) proposed a preference-based MAB called dueling bandits. Dudík et al. (2015) developed it into contextual dueling bandits, and Di et al. (2025) optimized contextual dueling bandits under adversarial feedback. Sui et al. (2018) provides an extensive survey for advancements in dueling bandits. Much of this literature assume a known user objectives and aims to identify a winner under standard solution concepts (e.g., a von Neumann winner). In our application, the user objectives are not defined and are unknown *a priori*; preferences arise from latent, hard-to-articulate objectives that must be inferred online.

While there is past work on latent representations for single arms (Zhou & Brunskill, 2016; Maillard & Mannor, 2014; Hong et al., 2020) and on preference bandits based on Thompson sampling approaches with asymptotic guarantees (Mwai et al., 2025), in this paper, we derive *HEART-UCB*, a latent dueling bandits algorithm, which is well-suited to short-horizon settings such as LLM selection. Further, unlike prior work that uses language primarily for sentiment analysis or context summarization in bandits (Boun-effouf & Feraud, 2025), we propose a principled way to incorporate language feedback into the UCB utility. To constrain cost, we reformulate knapsack-based bandit methods originally developed for non-dueling settings (Immor-lica et al., 2022; Badanidiyuru et al., 2018) to accommodate dueling feedback with both cost and round budgets.

In line with recent bandit research that advances the field through integration of complementary techniques (Di et al., 2025; Wang et al., 2025b), principled application to new domains (Li et al., 2010; Agarwal et al., 2024; Kong et al., 2024; Nguyen et al., 2025), or additive extension of classical methods (Badanidiyuru et al., 2018; Zhang & Cheung, 2024), our work contributes along all three axes: formulating latent dueling bandits with UCB, applying them to user-level LLM matching, and introducing bias, penalty, and cooldown mechanisms tailored to the application at hand.

## 3. Method

### 3.1. Problem setup

We study an interactive LLM-to-user matching problem under uncertainty and resource constraints. The setup consists of three components.

**LLM pool.** We are given a finite set of $M$ LLMs, $\mathcal{A} = \{a^{(m)}\}_{m=1}^M$. Each LLM possesses distinct latent capabilities (e.g., reasoning quality, verbosity, latency, cost efficiency, ability to process long context) that are not fully specified or observable *a priori*. Hence, neither the user nor our proposed algorithm, CUPID, has direct knowledge of the capabilities of an arbitrary LLM. This mirrors real-world deployment scenarios where practitioners typically lack a precise understanding of what an arbitrary LLM excels at and must discover its characteristics through use and comparison against alternatives.

**User preferences.** The user also has $K$ unknown, latent objectives, $\mathcal{Z}$, such as accuracy, cost, latency, and response style, when having preferences over LLMs. These preferences are subjective, application-dependent, and hard to explicitly articulate upfront. The user does not know which model best satisfies their objectives and can only express preferences implicitly through pairwise comparisons of model responses. We capture this interaction by sequentially collecting pairwise preference feedback over $T$ rounds, yielding a dataset, $\mathcal{D} = \{(a_t^{(u)} \succ a_t^{(v)})\}_{t=1}^T$, where $a_t^{(u)} \succ a_t^{(v)}$ indicates that the user preferred the response produced by LLM $a_t^{(u)}$ over LLM $a_t^{(v)}$ at round $t$, when evaluated on the same prompt.

We model the user preference via a latent function $f : \mathcal{A} \to \mathbb{R}$, where $f(a)$ represents the (unobserved) *utility* assigned by the user to LLM $a$. This formulation allows user preferences to be inferred from comparative feedback, without requiring explicit numerical ratings or objective-specific annotations. Optionally, the user may also provide lightweight natural language feedback at each round, indicating salient objectives or reasons underlying a particular preference. Such feedback, when available, will also be incorporated to

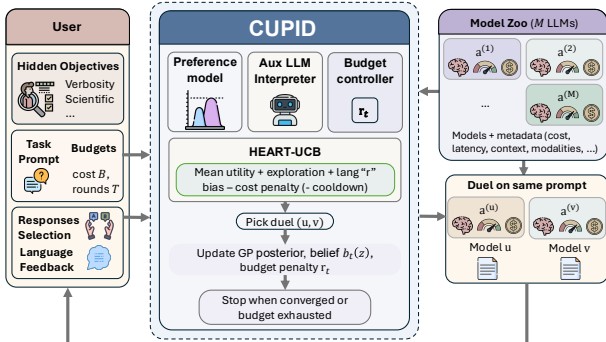

*Figure 2.* Based on *pairwise comparisons* of LLM responses from the *model zoo* and *optional language feedback* provided by a user with *hidden objectives*, a posterior distribution over model preferences and a latent belief is iteratively updated (red loop), while meeting user specified *cost and round budget constraints*.

supplement preference signals (Section 3.2.4).

**CUPID, the matchmaker.** CUPID progressively "matches" a user to an LLM by mediating interactions and comparisons, using minimal feedback to uncover which model best satisfies the user's implicit objectives. Its goal is to identify the LLM that best aligns with the user's latent preferences while respecting explicit resource constraints: a total monetary budget, $\$B$, and an interaction budget, $T$ rounds. At each round, CUPID must balance exploration of uncertain LLM capabilities with exploitation of inferred user preferences, using only lightweight user feedback.

### 3.2. HEART-Upper Confidence Bound

As the core algorithm of the CUPID framework, in this section, we develop a novel UCB, named Human-Elicited Adaptive Resource-aware Tradeoffs (HEART-UCB).

#### 3.2.1. PRELIMINARIES: UCB

Consider a bandit problem with a finite action set $\mathcal{A}$, where each action $a \in \mathcal{A}$ has an unknown expected reward (we reuse the notation $a$, because, in CUPID, $a$ = arm = LLM). At round $t$, the learner selects $a_t = \arg\max_{a \in \mathcal{A}} \text{UCB}_a(t)$, where $\text{UCB}_a(t) = \hat{\mu}_a(t) + \beta\,\hat{\sigma}_a(t)$, with the estimated mean reward, $\hat{\mu}_a(t)$, and an uncertainty measure, $\hat{\sigma}_a(t)$, and the exploration–exploitation trade-off parameter, $\beta > 0$.

#### 3.2.2. DUELING BANDITS FOR LLM RESPONSES

Because user utilities $\{f(a^{(m)})\}_{m=1}^M$ are latent and only observed indirectly through noisy pairwise comparisons, we require a probabilistic model that (i) represents uncertainty over all LLMs jointly, and (ii) generalizes preference information from observed comparisons to unqueried LLMs. Gaussian processes provide a natural nonparametric prior for this purpose, enabling uncertainty-aware exploration.

We place a Gaussian process prior (Rasmussen & Williams, 2005) over user preference utility $f \sim \mathcal{GP}(0, k(\cdot, \cdot))$. Equivalently, letting $\mathbf{f} = [f(a^{(1)}), f(a^{(2)}), \ldots, f(a^{(M)})]^\top$, we have the finite-dimensional Gaussian prior, $\mathbf{f} = \mathcal{N}(\mathbf{0}, \mathbf{\Sigma})$, of preferences over all LLMs, where $\mathbf{\Sigma}_{uv} = k(a^{(u)}, a^{(v)})$. The kernel, $k(\cdot, \cdot)$, captures similarity between LLMs in terms of their latent capability profiles, allowing preference information about one LLM to inform about others.

When two LLMs are queried and the user preference, $a_t^{(u)} \succ a_t^{(v)}$, at round $t$ is provided, the preference likelihood can be computed using a probit (Chu & Ghahramani, 2005),

$$P(a_t^{(u)} \succ a_t^{(v)} \mid \mathbf{f}) = \Phi\left(\frac{f(a_t^{(u)}) - f(a_t^{(v)})}{\sqrt{2}\sigma}\right) \quad (1)$$

where $\Phi(\cdot)$ is the standard Gaussian CDF and $\sigma^2$ captures observation noise (see Appendix A.1 for the rationale). Given the preference dataset $\mathcal{D}$, *our objective is to compute the posterior utility of each LLM so that, for any pair of LLMs, $a^{(r)}$ and $a^{(s)}$, both (i) the probability that $a^{(r)}$ is preferred over $a^{(s)}$ and (ii) our uncertainty in that estimate* (Appendix A.2). With this probit likelihood and the Gaussian process prior, the posterior prediction is intractable, hence requiring an approximate posterior estimation technique. Following Williams & Barber (1998) for Laplace approximation, we can obtain a closed-form, uncertainty-aware estimate of pairwise user preferences for any new pair of queries (Appendix A.4). However, *exhaustively comparing all $\mathcal{O}(M^2)$ LLM pairs is computationally and interaction-wise prohibitive* under limited user feedback and budget constraints. Instead, **the challenge is to strategically select which LLM comparisons to perform so as to efficiently identify the LLM that best aligns with the user's preferences**. This naturally leads to a sequential decision making formulation.

We treat *an LLM as an arm* with an unknown latent utility in a bandit setting. At each round $t$, we assign each LLM an upper confidence bound (UCB) score,

$$\text{UCB}_t^{(m)} = \hat{\mu}_t^{(m)} + \beta\,\hat{\sigma}_t^{(m)}, \quad \forall m \in \{1, \ldots, M\} \quad (2)$$

where $\hat{\mu}_t^{(m)}$ and $\hat{\sigma}_t^{(m)}$ denote the posterior mean and standard deviation of latent utility of LLM $a^{(m)}$, respectively, obtained from the marginal of the GP posterior over utilities and, $\beta > 0$ trades off exploitation of high-utility LLMs with exploration of uncertain ones.

### 3.2.3. LATENT DUELING BANDITS FOR MATCHMAKING

In practice, user preferences are rarely driven by a single, fixed objective. Instead, they often reflect a combination of multiple, implicit, hard-to-articulate *objectives (e.g., accuracy, verbosity, latency, cost)*, whose relative importance is

difficult to explicitly specify or articulate upfront and may vary across interactions. As a result, the latent utility of an LLM depends not only on the model itself, but also on an unobserved multi-objective state of the user. This requires (i) adapting the pairwise preference model to condition on latent objectives, and (ii) maintaining and updating a belief over these objectives during interaction.

**Latent preference model.** Let $\mathcal{Z} = \{z^{(k)}\}_{k=1}^K$ denote a set of latent user objectives. The elements in $\mathcal{Z}$ jointly represent user objectives such as cheap, verbose, etc., where each $z^{(k)}$ need not correspond to a physically interpretable objective in isolation. To model uncertainty over (unknown) these user objectives, at round $t$, we define a belief vector, $\mathbf{b}_t \in \Delta^{K-1}$, on the $K - 1$ probability simplex, where $b_t(z)$ denotes the relative weight assigned to objective $z$, and $\sum_{z \in \mathcal{Z}} b_t(z) = 1$. We define a latent utility function, $f : \mathcal{Z} \times \mathcal{A} \to \mathbb{R}$, where $f(z, a)$ represents the latent utility of LLM $a$ under objective $z$. By replacing $f(a_t)$ with $f(a_t, z_t)$ in (1) to condition on the objective vector $z$, we derive a posterior predictive as summarized in Theorem 3.1.

**Theorem 3.1** (Posterior Predictive Preference under Latent Objectives)**.** *Under the Gaussian process posterior over the latent utility function $f(z, a)$, the posterior predictive probability that LLM $a^{(r)}$ is preferred over LLM $a^{(s)}$, conditioned on a latent objective $z$ and preference data $\mathcal{D}_{1:t}$, satisfies,*

$$P(a^{(r)} \succ a^{(s)} \mid z, \mathcal{D}_{1:t}) = \Phi\left(\frac{\hat{\mu}_z^{(r)} - \hat{\mu}_z^{(s)}}{\hat{\sigma}_z}\right), \quad (3)$$

*where $\hat{\mu}_z^{(m)}$ and $\hat{\sigma}_z^2 = 2\sigma^2 + \hat{\Sigma}_z^{(rr)} + \hat{\Sigma}_z^{(ss)} - 2\hat{\Sigma}_z^{(rs)}$ denote the posterior mean and variance of the latent utility $f(z, a)$. Here, $\hat{\Sigma}$ is the posterior covariance matrix of the latent utility vector $\mathbf{f}$, obtained under the Laplace approximation and $\hat{\Sigma}^{(ij)}$ denotes its $(i, j)$-th entry.*

*Proof sketch.* Redefine the likelihood in (1) in terms of the joint latent utility $f(z, a)$. Applying Bayes' theorem yields the posterior distribution over $\mathbf{f}$, which is approximated using a Laplace approximation. See Appendix A.4. $\square$

**Belief over latent objectives.** After observing a comparison outcome $a_t^{(u)} \succ a_t^{(v)}$ at round $t$, we update the belief via Bayes' rule (in log domain for numerical stability):

$$\log b_{t+1}(z) \propto \log b_t(z) + \log P(a_t^{(u)} \succ a_t^{(v)} \mid z, \mathcal{D}_{1:t}), \quad (4)$$

then normalize with,
$\mathbb{1}_{a_t^{(u)} \succ a_t^{(v)}} \cdot P(a_t^{(u)} \succ a_t^{(v)} \mid z, \mathcal{D}_{1:t}) + \mathbb{1}_{a_t^{(v)} \succ a_t^{(u)}} \cdot P(a_t^{(v)} \succ a_t^{(u)} \mid z, \mathcal{D}_{1:t})$ with preference indicator $\mathbb{1}$.

Departing from vanilla UCB that assumes a fixed reward distribution, to select LLMs under uncertainty over both latent

utilities and user objectives, with $\hat{\mu}_z$ and $\hat{\Sigma}_z$ from Theorem 3.1, we obtain a belief-aware UCB (bUCB) for each LLM by marginalizing over all latent belief on objectives:

$$\text{bUCB}_t^{(m)} = \sum_{k=1}^{K} b_t(z^{(k)}) \left( \hat{\mu}_z^{(m)} + \beta \sqrt{\hat{\Sigma}_z^{(mm)}} \right). \quad (5)$$

This formulation generalizes dueling bandits to a latent multi-objective setting, where user preferences arise from an unknown and evolving combination of objectives. By jointly reasoning over uncertainty in both latent utilities and objective states, the algorithm adaptively selects informative comparisons and efficiently identifies LLMs that best align with the user's preferences (see Algorithm 1).

### 3.2.4. MODULATING LANGUAGE FEEDBACK

In certain cases (e.g., LLM provider-published metadata, model cards, benchmarks, user studies, etc.), we have access to a finite set of potential objectives or attributes (e.g., cost, latency, verbosity, safety, domain specialization), even though the user's true preference over these objectives is unknown and difficult to articulate precisely. In such cases, *natural language feedback can convey directional intent*, highlighting which subset of objectives should be emphasized. Our rationale is inspired by findings in psychology (Tversky & Shafir, 1992), which show that when alternatives trade off different attributes and are similarly attractive, decision makers often defer choice and seek additional information. In such settings, language feedback provides such directional information, allowing CUPID to bias exploration toward relevant LLMs while leveraging previously learned pairwise preferences.

At round $t$, the user may optionally provide a brief natural language cue (e.g., "I need a cheap model"). As illustrated in Appendix A.7, an auxiliary LLM (we choose Grok 4.1 Fast for its semantics capabilities (xAI, 2025), see Appendix C.2 for auxiliary LLM comparison) is prompted with this feedback together with model metadata and produces a set of arm-level indicator functions, $\{\mathbb{1}_{a_t^{(m)}} \in \{0,1\}\}_{m=1}^{M}$, that signals if the LLM $a_t^{(m)}$ aligns with the direction expressed in the user's language feedback at round $t$.

A scaled indicator function, $l_t^{(m)} \mathbb{1}_{a_t^{(m)}}$, is then added as bias to $\text{bUCB}_t^{(m)}$ updates in (5). We define this scaling factor as, $l_t^{(m)} = \max\left( \min(\lambda \text{SD}_t, \overline{\lambda} \text{SD}_t), \underline{\lambda} \right)$, where $\text{SD}_t$ is the standard deviation of $\{\text{bUCB}_t^{(m)}\}_{m=1}^{M}$ in (5) that provides a robust measure of bias dispersion, ensuring that the influence of language feedback adapts to how well-separated models currently are. $\lambda$ controls the strength of language influence, $\overline{\lambda}$ enforces a safety cap, and $\underline{\lambda}$ ensures a minimum nonzero effect. The rationale accompanied by an illustration of this computation is provided in Appendix A.7.

---

**Algorithm 1** CUPID

**Require:** LLM pool $\{a^{(m)}\}_{m=1}^{M}$, budget $\$B$, rounds $T$
    Initialize latent belief $b_0(z)$
1: **for** $t = 1$ **to** $T$ **do**
2:    $\hat{\mu}_t, \hat{\Sigma}_t \leftarrow$ GP posterior $(\mathcal{D}_{1:t-1}, b_{t-1}(z))$
3:    HEART-UCB$_t^{(m)} \leftarrow$ Update $\left( \hat{\mu}_t, \hat{\Sigma}_t, b_{t-1}(z), x_{t-1} \right)$
4:    $a^{(u)}, a^{(v)} \leftarrow$ TopArms $\left( \{\text{HEART-UCB}_t^{(m)}\}_{m=1}^{M} \right)$
5:    $y^{(u)} \leftarrow$ LLM$(a^{(u)}); y^{(v)} \leftarrow$ LLM$(a^{(v)})$
6:    $\mathcal{D}_t \leftarrow \mathcal{D}_{t-1} \cup Is\left( a^{(u)} \succ a^{(v)}; y^{(u)}, y^{(v)} \right)$
7:    $x_t \leftarrow$ UserLang$(y^{(u)}, y^{(v)})$ (optional language)
8:    $b_t(z) \leftarrow$ BeliefUpdate $(b_{t-1}(z))$
9: **end for**

---

### 3.2.5. ENFORCING BUDGET CONSTRAINTS

To strictly control resource consumption (maximum monetary budget $B$ and interaction rounds $T$) while maximizing utility, we adopt a time-varying resource penalty, $r_t$, which increases when the algorithm spends budget faster than a target rate. By adding a cost sensitivity parameter, $\tau$ to the per-round expenditure, $B/T$, the *effective per-round expenditure* is defined as $c_{\text{target}} = B/(T + \tau)$, where $T + \tau > 0$. The parameter $\tau$ allows the user to adjust the pacing of the budget: $\tau > 0$ tightens the budget to conserve resources for later rounds (conservative) and $\tau < 0$ encourages earlier exploration of expensive models to speed up convergence (aggressive). $\tau$ can also be a nonstationary variable.

At each round $t$, after selecting a pair of models and incurring a cost $c_t$, we update the resource penalty as $r_{t+1} = \max(0, r_t + \eta (c_t - c_{\text{target}}))$ for $\eta > 0$. This penalty $r_t$ acts as a conversion factor between cost and utility. When scoring a candidate pair of LLMs with provider given input-output cost, $C^{(m)}$, we subtract the penalty $C^{(m)} \cdot r_t$ from their bUCB scores. As a result, if the system overspends $(c_t > c_{\text{target}})$, $r_t$ grows, disproportionately penalizing expensive pairs. This forces the algorithm to shift exploration toward cheaper alternatives until the average spending realigns with $c_{\text{target}}$. Conversely, if spending is low, $r_t$ decays to zero, allowing the algorithm to freely select high-utility models regardless of cost.

### 3.2.6. HEART-UCB UPDATE

By incorporating bias terms in Sections 3.2.4 and 3.2.5 into (5) to *jointly optimize* the latent utility, language alignment, and cost, we propose Human-Elicited Adaptive Resource-aware Tradeoffs HEART-UCB$_t^{(m)}$

$$= \sum_{k=1}^{K} \underbrace{b_t(z^{(k)})}_{\text{belief}} \Big( \underbrace{\hat{\mu}_t(z)}_{\text{expected utility}} + \underbrace{\beta_0 \sqrt{\hat{\Sigma}_z^{(mm)}}}_{\text{exploration bonus}} \Big)$$
$$\underbrace{\phantom{\sum_{k=1}^{K} b_t(z^{(k)}) \Big( \hat{\mu}_t(z) + \beta_0 \sqrt{\hat{\Sigma}_z^{(mm)}} \Big)}}_{\text{user objective-aware UCB}} \qquad (6)$$
$$+ \underbrace{\beta_1 l_t^{(m)} \mathbb{1}_{a_t^{(m)}}}_{\text{language-driven bias}} - \underbrace{\beta_2 C^{(m)} r_t}_{\text{resource penalty}} - \underbrace{\beta_3 w_t^{(m)}}_{\text{cooldown}},$$

where $\beta_0, \beta_1, \beta_2$, and $\beta_3$ are scaling factors, and the optional cooldown term $w_t^{(m)}$ penalizes arms that have been selected frequently in recent rounds, encouraging diversity and preventing the algorithm from repeatedly querying the same LLM when several candidates have comparable utility. HEART-UCB is our main algorithmic contribution and, to the best of our knowledge, extends standard UCB in two key ways: by formulating latent dueling bandits and by incorporating language-driven UCB bias shaping.

### 3.3. Effect of Language Feedback on Preference.

Language feedback in HEART-UCB enters as an additive bias on the latent utility and induces a structured deformation of pairwise preference boundaries, rather than introducing uncontrolled nonstationarity (Figure 6). For any context $z \in \mathcal{Z}$ and arms $u, v \in \mathcal{A}$, we define the language-shaped utility $g_\star(z, u) = f_\star(z, u) + \gamma B(u)$, which modifies the probit comparison probability to

$$\mathbb{P}(a_u \succ a_v \mid z) = \Phi\left( \frac{f_\star(z, u) - f_\star(z, v) + \gamma \Delta B(u, v)}{\sqrt{2}} \right),$$

with $\Delta B(u, v) = B(u) - B(v) = \beta_1 l_t^{(m)} \mathbb{1}_{a_t^{(m)}}$. As a result, the $\frac{1}{2}$-probability decision boundary is translated by $\gamma \Delta B(u, v)$ in latent utility space. This translation is monotone in the feedback bias and uniformly bounded, implying that language feedback shifts preferences in a predictable and controlled manner without overpowering the intrinsic utility signal.

When budget constraints are incorporated, the effective shaped utility becomes $h_\star(z, u) = f_\star(z, u) + \gamma B(u) - \kappa c(u)$, leading to the tie condition $f_\star(z, u) - f_\star(z, v) + \gamma \Delta B(u, v) - \kappa \Delta c(u, v) = 0$, where $\Delta c(u, v) = c(u) - c(v)$. This linear relation reveals an explicit exchange rate between qualitative guidance and quantitative cost: holding intrinsic utilities fixed, one unit of relative language bias offsets $\gamma / \kappa$ units of additional cost. Language feedback and budget constraints interact additively and transparently in the latent space. Language feedback functions as a bounded shaping signal that steers exploration toward user-preferred regions of the model space while preserving asymptotic efficiency and robustness. We provide a regret bound analysis in Appendix A.8

## 4. Experiments

In this section, we evaluate the performance of CUPID in terms of alignment with user objectives and cost effectiveness for both text and image generation, capping the maximum cost incurred by the authors for LLM API calls per experiment at \$75 (i.e., for image models). As **benchmarks**, we compare against a latent dueling bandit setting, $\epsilon$-greedy dueling bandits (Yue et al., 2012; Yue & Joachims, 2009) and two Bradley–Terry–endowed dueling baselines: RUCB (Zoghi et al., 2014) and LMA, a variant of LMArena in (Chiang et al., 2024; LMArena, 2025) (see Appendix D.3). Since the latter does not impose any budget constraints, it is expected to identify the LLM that best matches user preferences when cost is ignored.

### 4.1. Automated Performance Validation

**Setup.** We first consider a representative real-world scenario in which a user selects a desired LLM from a model zoo. A key requirement for benchmarking is access to a model zoo with pre-specified objectives. We therefore consider a collection of 25 LLMs from OpenAI, each annotated with $K' = 34$ objectives and a fixed budget (see Appendix C.1). We evaluate four experimental setups (A1–A4 in Table 1), corresponding to four user types that differ in the number of assigned objectives, $K \subseteq K'$, relevant to the user (Appendix D.3). Each setup is repeated five times, yielding a total of 20 simulated users. To avoid information leakage, CUPID does not observe names of LLMs, and operates solely on objective-level metadata. Since human innate objectives are difficult to directly model, we rely on quantifiable objectives available in the dataset.

**Simulating users.** To enable automated evaluation, we simulate human users using an LLM-as-a-judge framework. Specifically, by employing prompt-engineering techniques described in Appendix B.1 and Appendix D.2.1, the judge has access to the ground-truth set of 25 LLM metadata and their full set of $K'$ objectives, as well as the subset of objectives $K$ associated with the simulated user type. At each round, the judge observes the LLM selected in the previous round, determines the preferred LLM in the current duel, and generates corresponding natural language feedback using another LLM (e.g., "I want a cheaper model") that reflects desired improvements over the previous choice.

**Results.** We make the following conclusions based on Table 1 and Figure 3:
*1. Ablations show incorporating a reasonable amount of language feedback can further enhance performance and cost efficiency.* In addition to CUPID, we also ablate the language and cost penalty components of (6). As shown in Figure 3, CUPID without language attains lower cost, whereas CUPID without cost penalty results in faster

*Table 1.* Total cost, convergence round (CR), and convergence percentage (CP) for multi-objective preference experiments on OpenAI models. Red boxes indicate failure to converge on all experiments, bold values indicates the best performance, and underlined values indicates the second best performance. $K$ is the number of latent objectives and $M^*$ is the number of LLMs that are equally good at satisfying the $K$ objectives.

| Algorithms | Setup A1 ($K = 5, M^* = 2$) | | | Setup A2 ($K = 7, M^* = 10$) | | | Setup A3 ($K = 9, M^* = 2$) | | | Setup A4 ($K = 5, M^* = 5$) | | |
|---|---|---|---|---|---|---|---|---|---|---|---|---|
| | Cost ($) ↓ | CR ↓ | CP ↑ | Cost ($) ↓ | CR ↓ | CP ↑ | Cost ($) ↓ | CR ↓ | CP ↑ | Cost ($) ↓ | CR ↓ | CP ↑ |
| Latent Dueling Bandits | $0.108 ± 0.016 | N/A | 0% | $0.049 ± 0.002 | 1 | 100% | $0.125 ± 0.006 | N/A | 0% | $0.122 ± 0.018 | 6.75 | 80% |
| RUCB | $0.161 ± 0.034 | N/A | 0% | $0.116 ± 0.033 | 1 | 33.3% | $0.117 ± 0.036 | 1 | 33.3% | $0.118 ± 0.038 | 1 | 33.3% |
| LMA | $0.115 ± 0.057 | 5 | 60% | $0.046 ± 0.026 | 2.6 | 100% | $0.125 ± 0.066 | 1 | 20% | $0.073 ± 0.061 | 3.5 | 80% |
| Cost-constrained LMA | $0.091 ± 0.055 | 5.75 | 80% | $0.042 ± 0.022 | 2.6 | 100% | $0.097 ± 0.062 | 1 | 20% | $0.052 ± 0.044 | 3 | 80% |
| $\epsilon$-greedy | **$0.055 ± 0.062** | 2.5 | 80% | $0.053 ± 0.058 | 1.8 | 100% | $\underline{0.058 ± 0.063}$ | 2.5 | 80% | $0.075 ± 0.079 | 2 | 80% |
| CUPID without cost penalty | $0.081 ± 0.033 | 4.2 | 100% | $\underline{0.037 ± 0.014}$ | 1 | 100% | $0.066 ± 0.050 | 2 | 100% | $0.091 ± 0.027 | 3 | 100% |
| CUPID without language | $0.060 ± 0.005 | 4.2 | 100% | $0.040 ± 0.020 | 1.6 | 100% | **$0.022 ± 0.027** | 2 | 100% | **$0.058 ± 0.013** | 3.4 | 100% |
| **CUPID** | $\underline{0.056 ± 0.016}$ | 2.2 | 100% | **$0.035 ± 0.014** | 1 | 100% | $0.070 ± 0.003 | 2 | 100% | $\underline{0.063 ± 0.012}$ | 2.4 | 100% |

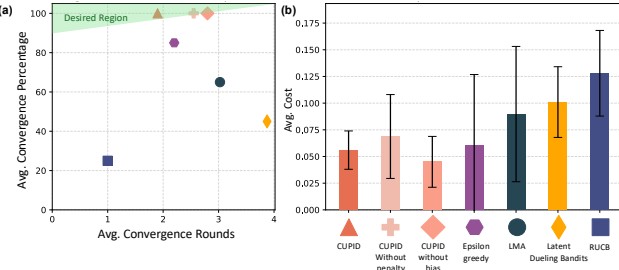

*Figure 3.* Convergence rounds and convergence percentages (left) and mean cost and standard deviation across all algorithms. All CUPID variants attained the maximum CP while converging at competitive cost and performance.

convergence but higher cost. Overall, CUPID provides the strongest balance of cost and convergence across all experiments.

2. *CUPID achieves reliable convergence across all experiments.* Across A1–A4 in Table 1, CUPID and both ablations converge in every run (CP = 100%). In contrast, the baselines exhibit frequent non-convergence: RUCB and latent dueling bandits fail in most settings (CP = 0%–33.3%), while LMArena and $\epsilon$-greedy show inconsistency.

3. *CUPID offers cost-efficiency.* CUPID is cost-efficient and stable with minimal cost variance across all setups, a property that is even more critical in high-cost LLM applications such as video generation, agentic systems with tools, and world models.

### 4.2. Robustness and Cross-provider Analysis

**Setup.** We consider a zoo of 37 LLMs from Artificial Analysis (LLM details in Appendix F.1) on the **MMLU-Pro** (Wang et al., 2024) and **AIME 2025** benchmarks (Artificial Analysis, 2026a;b; Lin, 2025).

For our robustness experiments, the best performing LLM in our dataset from AIME 2025 is designated as the ground

truth math expert (96.7% accuracy). LLM performance on AIME 2025 is normalized to $[0, 100]$. For each benchmark algorithm, we conduct ten rounds using the first ten prompts from the AIME 2025 dataset. Performance is defined as the normalized score of the final selected model. The procedure terminates early if the best model is identified.

To verify CUPID's performance over a range of diverse providers in a competitive setting, we further use the same protocol as Experiment 4.1 over 17 models with the highest MMLU Pro score, and use the best one as the ground truth.

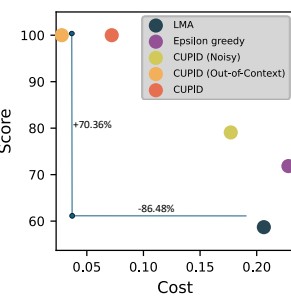

*Figure 4.* Robustness

**Robustness to noisy language feedback.** As detailed in Appendix F.2, we consider three types of noisy feedback: (i) *adversarial feedback*, (ii) *static feedback*, (iii) *gibberish feedback*. As shown in Figure 4, relative to LMArena, CUPID increases the score by 20.38 (+34.7%) while lowering cost by $0.0292 (−14.2%). $\epsilon$-greedy attains a higher score than LMArena (71.84) but does so at increased cost by +10.5%. In contrast, CUPID with noisy feedback outperforms $\epsilon$-greedy on both objectives, improving score by +7.24 while being 22.3% cheaper.

**Robustness to out-of-context objectives.** To evaluate robustness under objective shift, we decouple training objectives from evaluation objectives. Specifically, we use MMLU-Pro to provide objective-level signals reflecting overall model quality across *diverse* subjects, and then evaluate performance on mathematical reasoning *specific* using AIME-2025. Despite this mismatch, CUPID achieves the highest score (100.0) at the lowest cost ($0.0279). This corresponds to an 86.5% and 87.8% reduction in cost relative to LMArena and $\epsilon$-greedy, respectively, while simultaneously

improving scores by $+41.3$ and $+28.2$ points, respectively.

**Cross-provider experiment.** To simulate real world scenario, we run an experiment with a diverse providers model zoo with competitive-performance models.

*Table 2.* Total cost, convergence round (CR), and convergence percentage (CP) on 17 models over diverse providers.

| Method | Cost ($) | CR $\downarrow$ | CP $\uparrow$ |
|---|---|---|---|
| **CUPID** | $0.031 \pm 0.005$ | 3.2 | 100% |
| LMA | $0.045 \pm 0.023$ | 1 | 20% |
| $\varepsilon$-greedy | $0.026 \pm 0.014$ | 3.75 | 80% |

CUPID achieves the highest convergence percentage (100%) with the lowest variance. The heightened cost can be accounted for by the routing model, which takes roughly 30% of the total cost (Appendix C.2). Together with the robustness experiments, these results demonstrate CUPID crossprovider generalization capabilities.

### 4.3. Scalability and practical runtime

We expanded the model zoo to $M = 125$ models and shuffled it. Details at C.3.

*Table 3.* Scalability with the number of models. LLM indicates the use of auxiliary LLM. V2 (M = 75) consists of the first 75 models, V3 (M = 25) uses the OpenAI model zoo, and V4 (M = 5) consists of the last 5 models.

| Practical time in seconds | V1 (125) | V2 (75) | V3 (25) | V4 (5) |
|---|---|---|---|---|
| CUPID overhead | 0.719 | 0.267 | 0.094 | 0.035 |
| CUPID overhead (LLM) | 4.618 | 2.595 | 3.485 | 1.833 |
| Time-to-selection | 8.532 | 6.196 | 15.200 | 16.414 |
| Time-to-selection (LLM) | 12.431 | 8.524 | 18.591 | 18.212 |

**Results.** (1) CUPID's algorithmic overhead (GP + UCB + belief) scales with $M$ but remains <1 second even at M = 125–negligible compared to LLM generation time. (2) Time-to-selection is dominated by LLM API latency and actually decreases at larger $M$ in our experiments (V1/V2 are faster than V3/V4), because the model pool happened to contain faster-responding models. (3) The auxiliary LLM adds 2 - 4 seconds per round regardless of $M$, confirming it does not introduce a scalability bottleneck.

### 4.4. Validating Human Preference

**Study overview.** We conduct two A/B-blinded, within-subject user studies for text generation and text-to-image generation to evaluate CUPID's ability to match users to an LLM under realistic, resource-limited selection. Each study consists of three phases: (i) *Battle rounds.* In each round, participants interact in parallel with two LLMs suggested by CUPID and LMA, respectively (Appendix E).

Participants submit a single prompt to both algorithms and indicate their preferred response within each LLM (A or B is preferred). After all battle rounds, CUPID and LMA each nominate their own winning LLM. (ii) *Evaluation rounds.* Participants submit new prompts and observe the responses generated by the two winning LLMs. (iii) *Rating phase.* Participants rate each winning LLM on a 1–5 Likert scale (higher is better) based on response quality (alignment) and budget efficiency. Throughout the study, all 51 *participants were blinded* to both the underlying LLM identities and the selection algorithm (CUPID or LMA).

We consider four *levels of abstractness*, where higher levels require a potentially larger number of latent objectives:
**Setup H1.** Participants are provided with 2–4 explicit objectives and can view LLM specifications at each round.
**Setup H2.** Participants self-identify a domain (e.g., math) and aim to select a domain-appropriate LLM. While constrained to the chosen domain, they have greater flexibility than in H1.
**Setup H3.** Participants freely determine their objectives, unknown to the selection algorithm, based purely on subjective latent objectives.
**Setup H4.** Similar to H3, but for text-to-image generation.

**Results.** By construction, LMA is expected to select the highest-quality model, as it ignores cost considerations. In contrast, CUPID explicitly trades off quality and cost and may therefore select lower-cost models. Nonetheless, as shown in Figure 5(a), user ratings indicate that *CUPID achieves clear cost reductions while also maintaining comparable, or even better, model quality relative to LMA.* This effect becomes increasingly pronounced from H1 through H4. In simpler settings with only a few well-specified objectives (H1), CUPID can be redundant, although it still yields cost savings. As objective abstractness increases and user objectives become more latent (H2–H4), the need to infer hidden objectives grows, and CUPID's advantage becomes more evident, consistently delivering strong quality–cost trade-offs. Appendix E includes additional analysis.

## 5. Discussions

### 5.1. The Cold Start Problem

Bandit problems are generally susceptible to cold start problems. In round 1, the GP posterior equals the prior: all models have zero mean utility and equal variance under a stationary kernel. Selection is driven by (a) the language bias (if provided) and (b) the cost penalty. To mitigate this, we require an initial direction from the user, preventing over-exploration. Each pairwise outcome triggers two simultaneous updates: (1) the GP posterior and (2) the Bayesian belief over objectives. This dual update gives each sample a compounding effect, enabling faster convergence even from

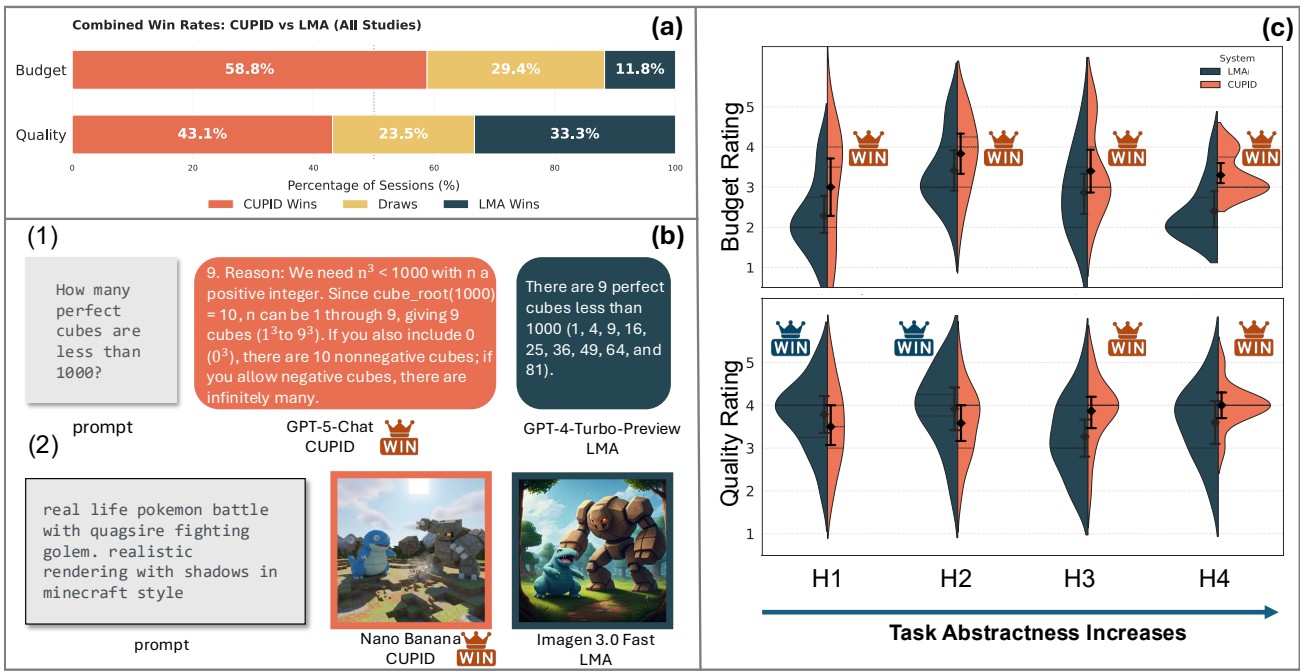

Figure 5. Human study results. (a) Combined win rates across all studies show CUPID outperforming the LMA baseline in both budget and quality. (b) Final matched model examples: (1) Text example; (2) Image example. (c) Rating results with different levels of abstractness.

uninformative priors.

## 5.2. Non-technical Users/High-stakes Domains

The assumption that user preferences reliably reflect true model quality is a general limitation of any human- or AI-driven feedback system, though it remains practical and informative in the absence of ground truth:

(1) CUPID's interaction is intentionally lightweight. The core mechanism is pairwise comparison ("Which response do you prefer?"), requiring no technical vocabulary. Optional language feedback is free-form, not structured.

(2) CUPID's advantage grows with task abstractness. Figure 5(c) shows CUPID's quality advantage increases from H1 (explicit) to H3/H4 (latent), suggesting greater benefit for non-technical users who cannot manually navigate model specifications.

(3) Domain verification is an open challenge. In high-stakes domains, CUPID could be extended with expert-validated test prompts or content-aware GP kernels (Section 5). This limitation applies equally to all preference-based methods.

Stratifying by self-reported AI familiarity, lower-familiarity participants (scores 2–3, where 1 = rarely used, 5 = daily use) achieved CUPID quality ratings of 3.67–3.63, comparable to or exceeding LMA ratings of 3.20–4.00 from higher-familiarity users (scores 4–5). This suggests that CUPID's guided exploration compensates for limited user expertise, enabling non-expert users to achieve competitive performance.

CUPID's modular architecture (Figure 2) allows adding safeguards without modifying the core HEART-UCB algorithm: filtering fluent-but-inaccurate models prior to selection via a hard constraint in the budget controller; adding calibration accuracy $z_{acc}$ into the belief vector $b_t$; adding static expert-ranked weights $S = [s_1, \ldots, s_M]$ to the language-driven bias every round.

## 6. Conclusions and Future Work

We introduced CUPID, a user-centric, budget-aware framework for matching users to LLMs under latent and difficult-to-articulate preferences. To this end, we derived a novel upper confidence bound named HEART-UCB. While our results demonstrate strong performance, the current Gaussian process takes into account the pair-wise preference from the user but does not automatically analyze the content of the response. A promising direction for future work is to learn response content-aware embeddings and incorporate them into the GP kernel to further accelerate convergence. Another direction is to incorporate informative priors derived from continuously evolving benchmarks and evaluation suites to improve cold-start performance. Sensitivity to prior and kernel specification also remains an important direction for future work. Finally, scaling our interface to a larger online participant pool would enable the first benchmark for user-level LLM matching.

## Impact Statement

This work aims to improve how users select LLMs by reducing unnecessary cost, interaction overhead, and reliance on poorly matched models. By enabling budget-aware and preference-driven model selection using lightweight feedback, CUPID can help individuals and organizations make more informed and efficient deployment decisions, potentially lowering financial and environmental costs associated with LLM usage.

CUPID does not introduce new generative capabilities or alter model outputs, but instead focuses on mediating access to existing models. As such, its primary societal impact lies in improving usability, transparency, and efficiency rather than amplifying risks related to content generation. Potential misuse, such as optimizing solely for cost at the expense of safety, can be mitigated through appropriate constraints and responsible deployment practices.

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

# A. Theoretical Analysis and Derivations

## A.1. Why probit?

Several likelihood models have been proposed for preference learning, including the Bradley–Terry (logistic) model, temperature-scaled variants thereof, and multi-way choice models such as Plackett–Luce. Bradley–Terry–type formulations are particularly effective in non-Bayesian or optimization-driven settings (e.g., gradient-based), but they primarily yield point estimates (e.g., MAP utilities) and provide limited support for principled posterior uncertainty estimation. Temperature-scaled variants introduce additional flexibility but rely on heuristic tuning and lack a clear probabilistic interpretation. Plackett–Luce models are well suited for multi-way ranking but impose stronger structural assumptions and are unnecessary in pairwise dueling settings. In contrast, the probit likelihood arises naturally from a Gaussian latent utility model and integrates cleanly with Gaussian process priors, enabling well-calibrated posterior uncertainty estimates that are critical for short-horizon exploration and reliable decision-making in our setting.

## A.2. Bayesian Update for GP Preference Learning

Let $\mathcal{A} = \{a^{(1)}, \ldots, a^{(M)}\}$ be the model set and $f \in \mathbb{R}^M$ be the latent utility vector with entries $f_m := f(a^{(m)})$. We place a GP prior $f \sim \mathcal{N}(0, \Sigma)$ where $\Sigma_{ij} = k(a^{(i)}, a^{(j)})$.

At round $t$ we observe a comparison $(i_t, j_t, y_t)$ where $y_t = +1$ means $a^{(i_t)} \succ a^{(j_t)}$ and $y_t = -1$ means the reverse. Under a probit preference model with noise standard deviation $\sigma > 0$,

$$\Pr(y_t = +1 \mid f) = \Phi\left(\frac{f_{i_t} - f_{j_t}}{\sqrt{2}\,\sigma}\right), \tag{7}$$

$$\Pr(y_t = -1 \mid f) = \Phi\left(\frac{f_{j_t} - f_{i_t}}{\sqrt{2}\,\sigma}\right), \tag{8}$$

where $\Phi$ is the standard Gaussian CDF.

Given data $D_t := \{(i_k, j_k, y_k)\}_{k=1}^t$, Bayes' rule yields

$$p(f \mid D_t) \propto p(f) \prod_{k=1}^t p(y_k \mid f).$$

## A.3. Laplace Approximation and Marginals

Because the probit likelihood is non-Gaussian, $p(f \mid \mathcal{D}_t)$ is intractable. Let the (negative) log-posterior be

$$\mathcal{L}_t(f) := -\sum_{k=1}^t \log p(y_k \mid f) + \frac{1}{2} f^\top K^{-1} f + \text{const.} \tag{9}$$

Let $\hat{f}_t$ be the MAP estimator:

$$\hat{f}_t \in \arg\min_{f \in \mathbb{R}^M} \mathcal{L}_t(f). \tag{10}$$

Define the Hessian at the MAP:

$$H_t := \nabla^2 \mathcal{L}_t(\hat{f}_t) = K^{-1} + W_t, \qquad W_t := -\nabla^2\left(\sum_{k=1}^t \log p(y_k \mid f)\right)\Big|_{f=\hat{f}_t}. \tag{11}$$

The Laplace approximation is

$$p(f \mid \mathcal{D}_t) \approx \mathcal{N}(f \mid \hat{f}_t, \hat{\Sigma}_t), \qquad \hat{\Sigma}_t := H_t^{-1}. \tag{12}$$

Therefore, the marginal posterior for arm $m$ is Gaussian with

$$\hat{\mu}_{t,m} := \mathbb{E}[f_m \mid \mathcal{D}_t] \approx (\hat{f}_t)_m, \qquad \hat{\sigma}_{t,m}^2 := \text{Var}(f_m \mid \mathcal{D}_t) \approx (\hat{\Sigma}_t)_{mm}. \tag{13}$$

**1. Laplace Approximation of the Posterior.** Since the probit likelihood is non-Gaussian, the true posterior $p(\mathbf{f} \mid \mathcal{D})$ is analytically intractable. We apply the Laplace approximation to approximate the posterior as a multivariate Gaussian centered at the Maximum A Posteriori (MAP) estimate $\hat{\mathbf{f}}$:

$$p(\mathbf{f} \mid \mathcal{D}) \approx \mathcal{N}(\mathbf{f} \mid \hat{\boldsymbol{\mu}}, \hat{\Sigma}), \tag{14}$$

where $\hat{\boldsymbol{\mu}} = \hat{\mathbf{f}}_{\text{MAP}}$ and $\hat{\Sigma} = (\mathbf{K}^{-1} + \mathbf{W})^{-1}$ is the inverse Hessian of the negative log-posterior.

**2. Distribution of the Utility Difference.** Let $\delta = f(a^{(r)}) - f(a^{(s)})$ be the difference in latent utility between the two items. Since $\mathbf{f}$ is approximated as Gaussian, the linear transformation $\delta$ is also Gaussian distributed:

$$p(\delta \mid \mathcal{D}) = \mathcal{N}(\delta \mid \mu_\delta, \sigma_\delta^2), \tag{15}$$

where the mean and variance are given by:

$$\mu_\delta = \hat{\mu}_r - \hat{\mu}_s, \tag{16}$$

$$\sigma_\delta^2 = \hat{\Sigma}_{rr} + \hat{\Sigma}_{ss} - 2\hat{\Sigma}_{rs}. \tag{17}$$

**3. Solving the Convolution.** Substituting the probit likelihood $P(a^{(r)} \succ a^{(s)} \mid \mathbf{f}) = \Phi\left(\frac{f_r - f_s}{\sqrt{2}\sigma}\right)$ into the integral, we seek to solve:

$$P(a^{(r)} \succ a^{(s)} \mid \mathcal{D}) = \int_{-\infty}^{\infty} \Phi\left(\frac{\delta}{\sqrt{2}\sigma}\right) \mathcal{N}(\delta \mid \mu_\delta, \sigma_\delta^2) \, d\delta. \tag{18}$$

We utilize the standard identity for the convolution of a Gaussian Cumulative Distribution Function (CDF) with a Gaussian Probability Density Function (PDF):

$$\int \Phi\left(\frac{x}{\beta}\right) \mathcal{N}(x \mid \mu, \tau^2) \, dx = \Phi\left(\frac{\mu}{\sqrt{\beta^2 + \tau^2}}\right). \tag{19}$$

Matching terms, we have $x = \delta$, $\mu = \mu_\delta$, $\tau^2 = \sigma_\delta^2$, and $\beta = \sqrt{2}\sigma$. Substituting these into the identity yields:

$$P(a^{(r)} \succ a^{(s)} \mid \mathcal{D}) = \Phi\left(\frac{\hat{\mu}_r - \hat{\mu}_s}{\sqrt{(\sqrt{2}\sigma)^2 + \sigma_\delta^2}}\right). \tag{20}$$

Finally, expanding $\sigma_\delta^2$ gives the result in Theorem A.1:

$$P(a^{(r)} \succ a^{(s)} \mid \mathcal{D}) = \Phi\left(\frac{\hat{\mu}_r - \hat{\mu}_s}{\sqrt{2\sigma^2 + \hat{\Sigma}_{rr} + \hat{\Sigma}_{ss} - 2\hat{\Sigma}_{rs}}}\right). \tag{21}$$

**A.4. Proof of Theorem 3.1**

**Theorem A.1** (GP preference prediction (Chu & Ghahramani, 2005)). *The GP prior and the probit pairwise likelihood defined above, the posterior predictive probability that LLM $a^{(r)}$ is preferred over LLM $a^{(s)}$ satisfies,*

$$P(a^{(r)} \succ a^{(s)} \mid \mathcal{D}) = \Phi\left(\frac{\hat{\mu}^{(r)} - \hat{\mu}^{(s)}}{\hat{\sigma}}\right), \tag{22}$$

*where, for the LLM $a^{(m)}$, $\hat{\mu}_m$ denotes the posterior mean utility and $\hat{\sigma}^2 = 2\sigma^2 + \hat{\Sigma}^{(rr)} + \hat{\Sigma}^{(ss)} - 2\hat{\Sigma}^{(rs)}$. Here, $\hat{\Sigma}$ is the posterior covariance matrix of the latent utility vector $\mathbf{f}$, obtained under the Laplace approximation and $\hat{\Sigma}^{(ij)}$ denotes its $(i, j)$-th entry.*

**Theorem A.2** (GP preference prediction; restatement). *Under the Laplace posterior approximation $p(f \mid \mathcal{D}_t) \approx \mathcal{N}(\hat{f}_t, \hat{\Sigma}_t)$ and the probit preference likelihood, the posterior predictive probability satisfies*

$$\Pr(a^{(r)} \succ a^{(s)} \mid \mathcal{D}_t) = \Phi\left(\frac{\hat{\mu}_{t,r} - \hat{\mu}_{t,s}}{\sqrt{2\omega^2 + \hat{\Sigma}_{t,rr} + \hat{\Sigma}_{t,ss} - 2\hat{\Sigma}_{t,rs}}}\right). \tag{23}$$

*Proof.* By marginalization,

$$\Pr(a^{(r)} \succ a^{(s)} \mid \mathcal{D}_t) = \int \Pr(a^{(r)} \succ a^{(s)} \mid f) \, p(f \mid \mathcal{D}_t) \, df. \tag{24}$$

Define the utility difference $\Delta := f_r - f_s$. Under the Gaussian approximation, $\Delta$ is Gaussian with

$$\Delta \mid \mathcal{D}_t \sim \mathcal{N}\Big(\hat{\mu}_{t,r} - \hat{\mu}_{t,s}, \ \hat{\Sigma}_{t,rr} + \hat{\Sigma}_{t,ss} - 2\hat{\Sigma}_{t,rs}\Big).$$

Also, the probit likelihood gives $\Pr(a^{(r)} \succ a^{(s)} \mid f) = \Phi\big(\Delta/(\sqrt{2}\omega)\big)$. Therefore,

$$\Pr(a^{(r)} \succ a^{(s)} \mid \mathcal{D}_t) = \mathbb{E}\Big[\Phi\Big(\frac{\Delta}{\sqrt{2}\omega}\Big)\Big].$$

Apply Lemma A.1 with $X = \Delta$, $\mu = \hat{\mu}_{t,r} - \hat{\mu}_{t,s}$, $\sigma^2 = \hat{\Sigma}_{t,rr} + \hat{\Sigma}_{t,ss} - 2\hat{\Sigma}_{t,rs}$, and $\tau = \sqrt{2}\omega$. This yields the stated closed form. $\square$

### A.5. Proof of Theorem 3.2 (Laplace Approximation)

**Theorem A.3** (Posterior predictive preference under latent objectives; restatement). *Fix a latent objective $z$. Assume the Laplace posterior for the vector $(f(z, a^{(1)}), \dots, f(z, a^{(M)}))$ is $\mathcal{N}(\hat{f}_{t,z}, \hat{\Sigma}_{t,z})$. Then for any $r, s$,*

$$\Pr(a^{(r)} \succ a^{(s)} \mid z, \mathcal{D}_t) = \Phi\left(\frac{\hat{\mu}_t(z, a^{(r)}) - \hat{\mu}_t(z, a^{(s)})}{\sqrt{2\omega^2 + \hat{\Sigma}_{t,z}(r,r) + \hat{\Sigma}_{t,z}(s,s) - 2\hat{\Sigma}_{t,z}(r,s)}}\right), \tag{25}$$

*where $\hat{\mu}_t(z, a^{(m)}) := (\hat{f}_{t,z})_m$.*

*Proof.* Conditioning on $z$ reduces the model to the same probit GP-preference model applied to the function $a \mapsto f(z, a)$. The proof is identical to Theorem 3.1, using the Gaussian approximation $p(f_z \mid \mathcal{D}_t) \approx \mathcal{N}(\hat{f}_{t,z}, \hat{\Sigma}_{t,z})$ and forming $\Delta_z := f(z, a^{(r)}) - f(z, a^{(s)})$. $\square$

### A.6. Bayesian Update for the Objective Belief

Assume $Z$ is a finite set (or a discretization) and $b_t(z) := \Pr(z \mid \mathcal{D}_t)$. Let $y_t$ be the observed comparison outcome at round $t$. Then

$$b_{t+1}(z) = \Pr(z \mid \mathcal{D}_{t+1}) = \frac{\Pr(y_t \mid z, \mathcal{D}_t) \, b_t(z)}{\sum_{z' \in Z} \Pr(y_t \mid z', \mathcal{D}_t) \, b_t(z')}. \tag{26}$$

Taking logs gives

$$\log b_{t+1}(z) = \log b_t(z) + \log \Pr(y_t \mid z, \mathcal{D}_t) - \log\Big(\sum_{z'} \Pr(y_t \mid z', \mathcal{D}_t) b_t(z')\Big), \tag{27}$$

which matches Eq. (4) up to the explicit normalization term.

### A.7. Language Feedback Modulation

Figure 6 illustrates how language is computed.

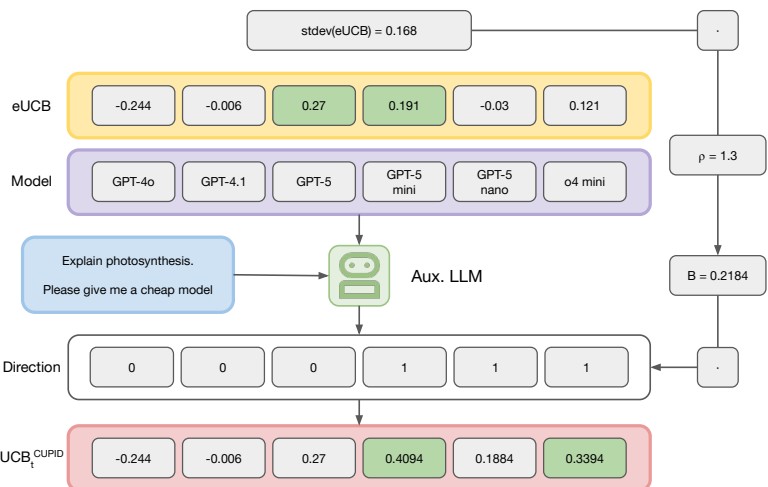

*Figure 6.* Language Feedback Modulation. The auxiliary model processes user inputs (blue) and the model information (purple). The bias is calculated and added to the initial eUCB (yellow), yielding an adjusted immediate bias for the subsequent turn (red). o4-mini, initially being excluded, attains higher score than the next GPT-5, an initial winner under the unbiased eUCB. In contrast, GPT-5 nano performs substantially weaker in prior rounds, resulting in its biased eUCB remaining lower than that of GPT-5.

## A.8. Regret Analysis

Assume $L_t(a) \leq f(a) \leq U_t(a)$ and selection $a_t = \arg\max_a [U_t(a) + \phi_t(a)]$. Define $\kappa_t = \phi_t(a^\star) - \max_{a \neq a^\star} \phi_t(a)$. Then

$$r_t \leq [U_t(a_t) - L_t(a_t) - \kappa_t]_+ \,.$$

For GP-UCB,

$$r_t \leq \left[ 2\sqrt{\beta_t}\sigma_{t-1}(a_t) - \kappa_t \right]_+ \,.$$

Thus, $\kappa_t > 0$ (informative) reduces regret, $\kappa_t = 0$ recovers standard UCB, and $\kappa_t < 0$ (misleading) increases regret only additively.

In the latent dueling setting, this becomes

$$r_t \leq \left[ 2\sqrt{\beta_t}\,\sigma_{t-1}(z^\star, a_t) - \kappa_t \right]_+ + 2\nu_t,$$

where

$$\nu_t = \sup_a \left| \sum_z b_t(z)\, U_t(z, a) - U_t(z^\star, a) \right|$$

(See Equation 5 notation).

Thus, informative language decreases regret, while latent-objective uncertainty contributes an additional vanishing term as the posterior over objectives concentrates.

This also yields the requested "exchange rate": the language margin $\kappa_t$ trades off directly against the exploration width $2\sqrt{\beta_t}\sigma_{t-1}(\cdot)$ with coefficient 1. Hence, one unit of informative language reduces the regret upper bound by one unit, while misleading language increases it by at most one unit (up to the positive-part clipping). In the latent setting, this holds up to the additional $2\nu_t$ uncertainty term.

Under a fixed-objective surrogate, language theoretically introduces an additive margin $\kappa_t$, improving cumulative regret by $-\sum_t \kappa_t$ while preserving the standard GP-UCB rate, and reducing sample complexity by $O\left(\frac{1}{\epsilon}\sum_{t=1}^{T}\kappa_t\right)$, as informative language eliminates suboptimal comparisons and accelerates learning, consistent with our empirical findings (Fig. 4, Table 1).

**Cumulative regret.** Starting from the instantaneous bound introduced in the rebuttal above, summing over $t = 1, \ldots, T$ gives

$$R_T = \sum_{t=1}^{T} r_t \leq \sum_{t=1}^{T} \left[2\sqrt{\beta_t}\,\sigma_{t-1}(\cdot) - \kappa_t\right]_- + +2\sum_{t=1}^{T}\nu_t.$$

Using standard GP-UCB analysis (Srinivas et al., 2010), and $\kappa_t^+ \leq 2\beta_t\sigma_{t-1}(\cdot)\forall t$, this yields

$$R_T = \tilde{O}(\sqrt{T\gamma_T}) - \sum_{t=1}^{T}\kappa_t + 2\sum_{t=1}^{T}\nu_t$$

Thus, informative language ($\kappa_t > 0$) reduces regret cumulatively, while misleading language ($\kappa_t < 0$) only affects it additively. $\kappa_t = 0$ recovers GP-UCB.

**Sample complexity/convergence.** The cumulative improvement directly implies faster convergence. In standard GP-UCB (Srinivas et al., 2010),

$$T_{\text{UCB}}(\epsilon) \sim \tilde{O}\left(\frac{\gamma_T}{\epsilon^2}\right).$$

With language,

$$T_{\text{HEART}}(\epsilon) \leq T_{\text{UCB}}(\epsilon) - O\left(\frac{1}{\epsilon}\sum_{t=1}^{T}\kappa_t\right),$$

showing that language feedback reduces the number of required interaction rounds by accelerating posterior concentration.

## B. Implementation Details

### B.1. LLM Prompting Strategy

We employ a multi-agent LLM architecture to facilitate user simulation and model selection. The prompts used for the "LLM-as-a-Judge," the "User Simulator," and the "Routing Model" are detailed below. To ensure consistent behavior, we use a temperature of 0.0 for all agents. Variable content injected at runtime is denoted by square brackets (e.g., `[Constraints]`).

#### B.1.1. LLM-AS-A-JUDGE

This agent evaluates which of two candidate models better satisfies the user's constraints. It is instructed to weigh trade-offs (e.g., speed vs. cost) similar to a human decision-maker.

**System Prompt: Judge Agent**

```
You are choosing between two language models for a user.  You are given:

   • A text description of the user's constraints.

   • A CSV with exactly two model rows (arm_A and arm_B).

Your task:  1) Read the constraints and the model metadata.  2) Decide which model
aligns better with the constraints.  All constraints are treated equally, and when
you choose a model, think about of the trade offs in that case before making the
decision.  For example trading 1 point of speed for $0.5 reduce in cost.  Try your
best to represent a human choosing between two options considering the trade offs.
3) Output exactly one of: 'arm_A' or 'arm_B'. Output no other words.
The two candidate models (in CSV form) are:  [Candidate Models CSV]
The constraint is:  [User Constraints]
```

### B.1.2. ROUTING MODEL

This agent filters the global pool of models based on the user's directional feedback, creating a candidate set for the bandit algorithm.

**System Prompt: Routing Agent**

```
IMPORTANT: Your role is to choose which language models align with the user's
preferences and DIRECTION.

You will receive:
- A free-form prompt that may contain a line 'DIRECTION: ...'  describing the user's
current preference direction (e.g.  wanting a cheaper model).
- Metadata for the last chosen model, so you can compare (e.g.  find models that are
cheaper, faster, etc.).
- A CSV table of all available models.

There are [N] models (rows) in the CSV, in fixed order.

Your only task:
1) For each model (each row), decide if it matches the user's preferences and
DIRECTION, possibly relative to the last chosen model.
2) Output exactly [N] integers (space-separated), one per row in the CSV, in the
same order.  Use '1' if the model is acceptable, '0' if it is not.
3) Output no other words.

Last chosen model (CSV; may be '(none yet'):  [Last Model Metadata]
CSV of ALL models:  [Full Model Zoo CSV]
```

### B.2. Hyperparameter Settings

*Table 4.* Hyperparameter Settings

| Hyperparameter | Feature | Setting |
|---|---|---|
| $\beta$ | Exploration Weight | 1.5 |
| $\eta$ | Budget Penalty Learning Rate | 0.5 |
| $\tau$ | Penalty Equilibration parameter | Adaptive to current round |
| $\lambda$ | Language-driven bias weight | 1.3 |
| $\overline{\lambda}$ | Language-driven bias cap | 1.6 |
| $\underline{\lambda}$ | Minimum language-driven bias | 0.01 |

## B.3. Compute + software environment

botorch==0.10.0

Google Colab, Python 3.12.12 (main, Oct 10 2025, 08:52:57) [GCC 11.4.0]

# C. Experiment - Baseline Performance

## C.1. The "Model Zoo" Dataset

The 25 OpenAI LLM and all the objective-level metadata are detailed in Table 5.

## C.2. Auxiliary LLM

We reran experiment A1 and A2, observed the behavior and recorded the cost of the auxiliary model. The results are detailed in Table 6.

## C.3. Scalability and Practical Runtime Experiment

**Dataset.** We extend the model zoo in Table 5 to 125 models and remove the metadata (Table 7). We run twenty five rounds over seeds 42–46 for each experiment.

*Table 7.* Scaled model zoo

| id | model-id | model |
|---|---|---|
| 1 | deepseek/deepseek-v3.2-speciale | DeepSeek V3.2 Speciale |
| 2 | z-ai/glm-4.7 | GLM-4.7 |
| 3 | moonshotai/kimi-k2-thinking | Kimi K2 Thinking |
| 4 | deepseek/deepseek-v3.2 | DeepSeek V3.2 |
| 5 | x-ai/grok-4-fast | Grok 4 Fast |
| 6 | x-ai/grok-4.1-fast | Grok 4.1 Fast |
| 7 | z-ai/glm-4.6 | GLM-4.6 |
| 8 | z-ai/glm-4.6v | GLM-4.6V |
| 9 | anthropic/claude-haiku-4.5 | Claude 4.5 Haiku |
| 10 | minimax/minimax-m2.1 | MiniMax-M2.1 |
| 11 | z-ai/glm-4.5-air | GLM-4.5-Air |
| 12 | minimax/minimax-m2 | MiniMax-M2 |
| 13 | z-ai/glm-4.5 | GLM-4.5 |
| 14 | z-ai/glm-4.5v | GLM-4.5V |
| 15 | google/gemini-2.5-flash | Gemini 2.5 Flash |
| 16 | moonshotai/kimi-k2-0905 | Kimi K2 0905 |
| 17 | google/gemini-2.5-flash-lite | Gemini 2.5 Flash-Lite |
| 18 | openai/gpt-4.1 | GPT-4.1 |
| 19 | openai/o4-mini | o4-mini |
| 20 | openai/gpt-3.5-turbo-16k | gpt-3.5-turbo-16k-0613 |
| 21 | openai/gpt-3.5-turbo-instruct | gpt-3.5-turbo-instruct |
| 22 | openai/gpt-3.5-turbo | GPT-3.5-turbo |
| 23 | openai/gpt-4-turbo-preview | GPT-4-Turbo-Preview |
| 24 | openai/gpt-4-turbo | GPT-4-Turbo |
| 25 | openai/gpt-4.1-mini | GPT-4.1-mini |
| 26 | openai/gpt-4.1-nano | GPT-4.1-nano |
| 27 | openai/gpt-4 | GPT-4 |
| 28 | openai/gpt-5.1-codex | GPT-5.1 Codex |
| 29 | openai/gpt-4o-mini-search-preview | GPT-4o-mini-Search-Preview |
| 30 | openai/gpt-4o-mini | GPT-4o-mini |
| 31 | openai/gpt-4o | GPT-4o |
| 32 | openai/gpt-5-chat | GPT-5-Chat |

| id | model-id | model |
|----|----------|-------|
| 33 | openai/gpt-5-mini | GPT-5-mini |
| 34 | openai/gpt-5-nano | GPT-5-nano |
| 35 | openai/gpt-5.1-chat | GPT-5.1-Chat |
| 36 | openai/gpt-5.1 | GPT-5.1 |
| 37 | openai/gpt-5 | GPT-5 |
| 38 | openai/o3-mini | o3-mini |
| 39 | openai/o3 | o3 |
| 40 | openai/gpt-5.2 | GPT-5.2 |
| 41 | openai/gpt-5.2-chat | GPT-5.2-Chat |
| 42 | mistralai/mistral-large-2512 | Mistral Large 3 |
| 43 | mistralai/devstral-2512 | Devstral 2 |
| 44 | mistralai/ministral-8b-2512 | Ministral 3 8B |
| 45 | mistralai/ministral-14b-2512 | Ministral 3 14B |
| 46 | google/gemma-3-27b-it | Gemma 3 27B Instruct |
| 47 | meta-llama/llama-4-maverick | Llama 4 Maverick |
| 48 | google/gemma-3-12b-it | Gemma 3 12B |
| 49 | meta-llama/llama-4-scout | Llama 4 Scout |
| 50 | google/gemma-3-4b-it | Gemma 3 4B |
| 51 | meta-llama/llama-3.3-70b-instruct | Llama 3.3 70B Instruct |
| 52 | meta-llama/llama-3.1-8b-instruct | Llama 3.1 8B Instruct |
| 53 | meta-llama/llama-3.1-70b-instruct | Llama 3.1 70B Instruct |
| 54 | meta-llama/llama-3.2-3b-instruct | Llama 3.2 3B Instruct |
| 55 | meta-llama/llama-3.2-1b-instruct | Llama 3.2 1B Instruct |
| 56 | qwen/qwen3.6-plus:free | Qwen3.6 Plus |
| 57 | xiaomi/mimo-v2-pro | MiMo-V2-Pro |
| 58 | z-ai/glm-5-turbo | GLM 5 Turbo |
| 59 | moonshotai/kimi-k2.5 | Kimi K2.5 |
| 60 | nvidia/nemotron-3-super-120b-a12b | Nemotron 3 Super |
| 61 | xiaomi/mimo-v2-omni | MiMo-V2-Omni |
| 62 | z-ai/glm-5 | GLM 5 |
| 63 | openai/gpt-5.4 | GPT-5.4 |
| 64 | anthropic/claude-sonnet-4.5 | Claude Sonnet 4.5 |
| 65 | xiaomi/mimo-v2-flash | MiMo-V2-Flash |
| 66 | qwen/qwen3-235b-a22b-2507 | Qwen3 235B A22B Instruct 2507 |
| 67 | mistralai/mistral-nemo | Mistral Nemo |
| 68 | deepseek/deepseek-chat-v3-0324 | DeepSeek V3 0324 |
| 69 | qwen/qwen3.5-flash-02-23 | Qwen3.5-Flash |
| 70 | openai/gpt-5.4-mini | GPT-5.4 Mini |
| 71 | deepseek/deepseek-chat-v3.1 | DeepSeek V3.1 |
| 72 | qwen/qwen3.5-9b | Qwen3.5-9B |
| 73 | anthropic/claude-sonnet-4 | Claude Sonnet 4 |
| 74 | stepfun/step-3.5-flash | Step 3.5 Flash |
| 75 | qwen/qwen3.5-397b-a17b | Qwen3.5 397B A17B |
| 76 | qwen/qwen3.5-plus-02-15 | Qwen3.5 Plus 2026-02-15 |
| 77 | mistralai/mistral-small-3.2-24b-instruct | Mistral Small 3.2 24B Instruct |
| 78 | z-ai/glm-4.7-flash | Z.ai: GLM 4.7 Flash |
| 79 | deepseek/deepseek-v3.1-terminus | DeepSeek V3.1 Terminus |
| 80 | deepseek/deepseek-chat | DeepSeek V3 |
| 81 | x-ai/grok-4.20 | Grok 4.20 |
| 82 | anthropic/claude-3.5-haiku | Claude 3.5 Haiku |
| 83 | nvidia/nemotron-3-nano-30b-a3b | Nemotron 3 Nano 30B A3B |
| 84 | minimax/minimax-m2.7 | MiniMax M2.7 |
| 85 | minimax/minimax-m2.5 | MiniMax M2.5 |

| id | model-id | model |
|----|----------|-------|
| 86 | minimax/minimax-m2-her | MiniMax M2-her |
| 87 | minimax/minimax-m1 | MiniMax M1 |
| 88 | minimax/minimax-01 | MiniMax-01 |
| 89 | mistralai/mistral-small-2603 | Mistral Small 4 |
| 90 | qwen/qwen3-32b | Qwen3 32B |
| 91 | moonshotai/kimi-k2 | Kimi K2 0711 |
| 92 | x-ai/grok-4 | Grok 4 |
| 93 | inception/mercury-2 | Mercury 2 |
| 94 | amazon/nova-lite-v1 | Nova Lite 1.0 |
| 95 | x-ai/grok-3-mini | Grok 3 Mini |
| 96 | qwen/qwen3-235b-a22b | Qwen3 235B A22B |
| 97 | qwen/qwen3-next-80b-a3b-instruct | Qwen3 Next 80B A3B Instruct |
| 98 | amazon/nova-micro-v1 | Nova Micro 1.0 |
| 99 | mistralai/mistral-medium-3.1 | Mistral Medium 3.1 |
| 100 | mistralai/devstral-small | Devstral Small 1.1 |
| 101 | x-ai/grok-3 | Grok 3 |
| 102 | mistralai/mistral-small-3.1-24b-instruct | Mistral Small 3.1 24B |
| 103 | amazon/nova-2-lite-v1 | Nova 2 Lite |
| 104 | mistralai/ministral-8b-2512 | Ministral 3 8B 2512 |
| 105 | qwen/qwen3-14b | Qwen3 14B |
| 106 | qwen/qwen3-max | Qwen3 Max |
| 107 | meta-llama/llama-guard-4-12b | Llama Guard 4 12B |
| 108 | qwen/qwen3-30b-a3b-thinking-2507 | Qwen3 30B A3B Thinking 2507 |
| 109 | qwen/qwen-plus-2025-07-28 | Qwen Plus 0728 |
| 110 | microsoft/phi-4 | Phi 4 |
| 111 | mistralai/mistral-large-2411 | Mistral Large 2411 |
| 112 | qwen/qwen3-max-thinking | Qwen3 Max Thinking |
| 113 | perplexity/sonar-pro | Sonar Pro |
| 114 | perplexity/sonar | Sonar |
| 115 | openai/o4-mini-high | o4 Mini High |
| 116 | nvidia/nemotron-nano-9b-v2 | Nemotron Nano 9B V2 |
| 117 | mistralai/mistral-large | Mistral Large |
| 118 | mistralai/mistral-7b-instruct-v0.1 | Mistral 7B Instruct v0.1 |
| 119 | mistralai/mistral-medium-3 | Mistral Medium 3 |
| 120 | mistralai/devstral-medium | Devstral Medium |
| 121 | deepseek/deepseek-r1-distill-qwen-32b | R1 Distill Qwen 32B |
| 122 | nvidia/llama-3.1-nemotron-70b-instruct | Llama 3.1 Nemotron 70B Instruct |
| 123 | mistralai/mixtral-8x22b-instruct | Mixtral 8x22B Instruct |
| 124 | mistralai/pixtral-large-2411 | Pixtral Large 2411 |
| 125 | nvidia/llama-3.1-nemotron-ultra-253b-v1 | Llama 3.1 Nemotron Ultra 253B v1 |

**Results.** We detail the results of each CUPID components in Table 8.

## C.4. Baseline Algorithms

### C.4.1. LMA

We use the **LMArena** (Chiang et al., 2024; LMArena, 2025) methodology as the first baseline for this experiment. The setup can be represented through three steps.

*Table 5.* OpenAI Model Pool

| id | model-id | model | Intelligence | Speed | Text In | Image In | Voice In | Video In | Text Out | Image Out | Audio Out | Video Out | Reasoning | Input Price | Cached Price | Output Price | Context Window | Max Output | Know. Cutoff | Comp. Endpt | Resp. Endpt | Assist. Endpt | Batch Endpt | Fine-Tune Endpt | Streaming | Func. Calling | Struct. Output | Fine-Tuning | Distillation | Pred. Outputs | Rate Lim (Free) | Rate Lim (T1) | Rate Lim (T2) | Rate Lim (T3) | Rate Lim (T4) | Rate Lim (T5) |
|---|---|---|---|---|---|---|---|---|---|---|---|---|---|---|---|---|---|---|---|---|---|---|---|---|---|---|---|---|---|---|---|---|---|---|---|---|
| 1 | openai/chatgpt-4o-latest | ChatGPT-4o | 3 | 3 | 1 | 1 | 0 | 0 | 1 | 0 | 0 | 0 | 0 | 5 | - | 15 | 128k | 16k | 23-Oct | 1 | 1 | 0 | 0 | 0 | 1 | 0 | 0 | 0 | 0 | 1 | - | 30k | 450k | 800k | 2M | 30M |
| 2 | openai/gpt-4.1 | GPT-4.1 | 4 | 3 | 1 | 1 | 0 | 0 | 1 | 0 | 0 | 0 | 0 | 2 | 0.5 | 8 | 1M | 32k | 24-Jun | 1 | 1 | 1 | 1 | 1 | 1 | 1 | 1 | 1 | 1 | 1 | - | 30k | 450k | 800k | 2M | 30M |
| 3 | openai/o4-mini | o4-mini | 4 | 3 | 1 | 1 | 0 | 0 | 1 | 0 | 0 | 0 | 1 | 1.1 | 0.28 | 4.4 | 200k | 100k | 24-Jun | 1 | 1 | 0 | 1 | 0 | 1 | 1 | 1 | 0 | 0 | 0 | - | 100k | 2M | 4M | 10M | 150M |
| 4 | openai/gpt-3.5-turbo-16k | GPT-3.5-16k | 1 | 2 | 1 | 0 | 0 | 0 | 1 | 0 | 0 | 0 | 0 | 3 | - | 4 | 16k | 4k | 21-Sep | 1 | 1 | 0 | 1 | 0 | 1 | 0 | 1 | 0 | 0 | 0 | - | 200k | 2M | 800k | 10M | 50M |
| 5 | gpt-3.5-turbo-instruct | GPT-3.5-Inst | 1 | 2 | 1 | 0 | 0 | 0 | 1 | 0 | 0 | 0 | 0 | 1.5 | - | 2 | 4k | 4k | 21-Sep | 1 | 1 | 0 | 0 | 0 | 0 | 0 | 0 | 0 | 0 | 0 | - | 200k | 2M | 800k | 10M | 50M |
| 6 | openai/gpt-3.5-turbo | GPT-3.5-Trb | 1 | 2 | 1 | 0 | 0 | 0 | 1 | 0 | 0 | 0 | 0 | 0.5 | - | 1.5 | 16k | 4k | 21-Sep | 1 | 1 | 1 | 1 | 0 | 1 | 1 | 1 | 0 | 0 | 0 | - | 200k | 2M | 4M | 10M | 50M |
| 7 | openai/gpt-4-turbo-prev | GPT-4-Prev | 2 | 3 | 1 | 1 | 0 | 0 | 1 | 0 | 0 | 0 | 0 | 10 | - | 30 | 128k | 4k | 23-Dec | 1 | 1 | 1 | 0 | 0 | 1 | 1 | 1 | 0 | 0 | 0 | - | 30k | 450k | 600k | 800k | 2M |
| 8 | openai/gpt-4-turbo | GPT-4-Turbo | 2 | 3 | 1 | 1 | 0 | 0 | 1 | 0 | 0 | 0 | 0 | 10 | - | 30 | 128k | 4k | 23-Dec | 1 | 1 | 1 | 1 | 0 | 1 | 1 | 1 | 0 | 0 | 0 | - | 30k | 450k | 600k | 800k | 2M |
| 9 | openai/gpt-4.1-mini | GPT-4.1-Mini | 3 | 4 | 1 | 1 | 0 | 0 | 1 | 0 | 0 | 0 | 0 | 0.4 | 0.1 | 1.6 | 1M | 32k | 24-Jun | 1 | 1 | 1 | 1 | 1 | 1 | 1 | 1 | 1 | 0 | 1 | 40k | 200k | 2M | 4M | 10M | 150M |
| 10 | openai/gpt-4.1-nano | GPT-4.1-Nano | 2 | 5 | 1 | 1 | 0 | 0 | 1 | 0 | 0 | 0 | 0 | 0.1 | 0.03 | 0.4 | 1M | 32k | 24-Jun | 1 | 1 | 1 | 1 | 1 | 1 | 1 | 1 | 1 | 0 | 1 | 40k | 200k | 2M | 4M | 10M | 150M |
| 11 | openai/gpt-4 | GPT-4 | 2 | 2 | 1 | 0 | 0 | 0 | 1 | 0 | 0 | 0 | 0 | 30 | 0 | 60 | 8k | 8k | 23-Dec | 1 | 1 | 1 | 0 | 0 | 1 | 1 | 1 | 0 | 0 | 0 | - | 10k | 40k | 80k | 300k | 1M |
| 12 | openai/gpt-5.1-codex | GPT-5.1-Code | 4 | 3 | 1 | 1 | 0 | 0 | 1 | 0 | 0 | 0 | 1 | 1.25 | 0.13 | 10 | 400k | 128k | 24-Sep | 1 | 1 | 1 | 1 | 1 | 1 | 1 | 1 | 0 | 0 | 0 | - | 500k | 1M | 2M | 4M | 40M |
| 13 | openai/gpt-4o-mini-search | GPT-4o-Search | 2 | 4 | 1 | 0 | 0 | 0 | 1 | 0 | 0 | 0 | 0 | 0.15 | - | 0.6 | 128k | 16k | 23-Oct | 0 | 1 | 0 | 0 | 0 | 1 | 0 | 1 | 0 | 0 | 0 | 40k | 200k | 2M | 4M | 10M | 150M |
| 14 | openai/gpt-4o-mini | GPT-4o-Mini | 2 | 4 | 1 | 1 | 0 | 0 | 1 | 0 | 0 | 0 | 0 | 0.15 | 0.08 | 0.6 | 128k | 16k | 23-Oct | 1 | 1 | 1 | 1 | 1 | 1 | 1 | 1 | 1 | 1 | 1 | 40k | 200k | 2M | 4M | 10M | 150M |
| 15 | openai/gpt-4o | GPT-4o | 3 | 3 | 1 | 1 | 0 | 0 | 1 | 0 | 0 | 0 | 0 | 2.5 | 1.25 | 10 | 128k | 16k | 23-Oct | 1 | 1 | 1 | 1 | 1 | 1 | 1 | 1 | 1 | 1 | 1 | - | 30k | 450k | 800k | 2M | 30M |
| 16 | openai/gpt-5-chat | GPT-5-Chat | 3 | 3 | 1 | 1 | 0 | 0 | 1 | 0 | 0 | 0 | 0 | 1.25 | 0.13 | 10 | 128k | 16k | 24-Sep | 1 | 1 | 1 | 1 | 0 | 1 | 1 | 1 | 0 | 0 | 0 | - | 30k | 450k | 800k | 2M | 30M |
| 17 | openai/gpt-5-mini | GPT-5-Mini | 3 | 4 | 1 | 1 | 0 | 0 | 1 | 0 | 0 | 0 | 1 | 0.25 | 0.03 | 2 | 400k | 128k | 24-May | 1 | 1 | 0 | 1 | 0 | 1 | 1 | 1 | 0 | 0 | 0 | - | 500k | 2M | 4M | 10M | 180M |
| 18 | openai/gpt-5-nano | GPT-5-Nano | 2 | 5 | 1 | 1 | 0 | 0 | 1 | 0 | 0 | 0 | 1 | 0.05 | 0.01 | 0.4 | 400k | 128k | 24-May | 1 | 1 | 0 | 1 | 0 | 1 | 1 | 1 | 0 | 0 | 0 | - | 200k | 2M | 4M | 10M | 180M |
| 19 | openai/gpt-5.1-chat | GPT-5.1-Chat | 3 | 3 | 1 | 1 | 0 | 0 | 1 | 0 | 0 | 0 | 0 | 1.25 | 0.13 | 10 | 128k | 16k | 24-Sep | 1 | 1 | 1 | 1 | 0 | 1 | 1 | 1 | 0 | 0 | 1 | - | 500k | 450k | 800k | 2M | 30M |
| 20 | openai/gpt-5.1 | GPT-5.1 | 4 | 4 | 1 | 1 | 0 | 0 | 1 | 0 | 0 | 0 | 1 | 1.25 | 0.13 | 10 | 400k | 128k | 24-Sep | 1 | 1 | 1 | 1 | 0 | 1 | 1 | 1 | 0 | 0 | 0 | - | 500k | 1M | 2M | 4M | 40M |
| 21 | openai/gpt-5 | GPT-5 | 4 | 4 | 1 | 1 | 0 | 0 | 1 | 0 | 0 | 0 | 1 | 1.25 | 0.13 | 10 | 400k | 128k | 24-Sep | 1 | 1 | 1 | 1 | 0 | 1 | 1 | 1 | 0 | 1 | 0 | - | 500k | 1M | 2M | 4M | 40M |
| 22 | openai/o3-mini | o3-mini | 4 | 3 | 1 | 0 | 0 | 0 | 1 | 0 | 0 | 0 | 1 | 1.1 | 0.55 | 4.4 | 200k | 100k | 23-Oct | 1 | 1 | 0 | 1 | 0 | 1 | 1 | 1 | 0 | 0 | 0 | - | 100k | 200k | 4M | 10M | 150M |
| 23 | openai/o3 | o3 | 5 | 1 | 1 | 1 | 0 | 0 | 1 | 0 | 0 | 0 | 1 | 2 | 0.5 | 8 | 200k | 100k | 24-Jun | 1 | 1 | 1 | 1 | 0 | 1 | 1 | 1 | 0 | 1 | 0 | - | 30k | 450k | 800k | 2M | 30M |
| 24 | openai/gpt-5.2 | GPT-5.2 | 4 | 4 | 1 | 1 | 0 | 0 | 1 | 0 | 0 | 0 | 1 | 1.75 | 0.18 | 14 | 400k | 128k | 25-Aug | 1 | 1 | 1 | 1 | 0 | 1 | 1 | 1 | 0 | 1 | 0 | - | 500k | 1M | 2M | 4M | 40M |
| 25 | openai/gpt-5.2-chat | GPT-5.2-Chat | 3 | 3 | 1 | 1 | 0 | 0 | 1 | 0 | 0 | 0 | 0 | 1.75 | 0.18 | 14 | 128k | 16k | 25-Aug | 1 | 1 | 0 | 0 | 0 | 1 | 1 | 1 | 0 | 0 | 0 | - | 30k | 450k | 800k | 2M | 40M |

*Table 6.* Comparison of auxiliary LLMs for CUPID. Instructions Followed: the LLM consistently generated a string of binary tokens in the correct format. Constraints Followed: the generated string abides to the constraints of the direction text. Cost/Round: the average cost per round for this model. Accuracy: the accuracy of CUPID with the auxiliary LLM.

| Attribute | Llama 3.3 70B | Qwen3 235B | Gemini 2.5 FL | Gemini 3 Flash Preview | o3 | Grok 4.1 Fast |
|---|---|---|---|---|---|---|
| Instructions followed | ✗ | ✗ | ✗ | ✓ | ✓ | ✓ |
| Constraints followed | ✗ | ✗ | ✗ | ✓ | ✓ | ✓ |
| Cost/round | Open-source | Open-source | $0.00033 | $0.00201 | $0.00871 | $0.00191 |
| CUPID accuracy | 100% | 100% | 100% | 100% | 100% | 100% |

*Table 8.* Detailed breakdown of each component of CUPID.

| Stage | V1 (M = 125) | V2 (M = 75) | V3 (M = 25) | V4 (M = 5) |
|---|---|---|---|---|
| GP update | 0.172 | 0.084 | 0.052 | 0.023 |
| UCB scoring | 0.547 | 0.183 | 0.042 | 0.012 |
| Belief update | <0.0001 | <0.0001 | <0.0001 | <0.0001 |
| LLM duel calls | 7.813 | 5.929 | 15.106 | 16.379 |
| Auxiliary LLM (Gemini 3 Flash Preview) | 3.899 | 2.328 | 3.391 | 1.798 |

**Anchor model sampling.** Let $\theta_t \in \mathbb{R}^K$ denote the current Bradley–Terry (BT) scores, and let $n_k$ be the number of comparisons involving model $k$. We sample an anchor model $i$ from the distribution

$$p_i \propto \exp(\alpha_{\text{score}} \theta_{t,i}) \left(1 + \frac{\alpha_{\text{unc}}}{n_i + 1}\right),$$

so that higher-scoring and more uncertain models are selected more frequently. This mirrors the sampling strategy used in Chatbot Arena, where models with high scores or high uncertainty are surfaced more often to accelerate convergence of the leaderboard.

**Opponent sampling.** Conditional on the anchor model $i$, we sample an opponent $j \neq i$ from the distribution

$$q_j \propto \exp(\alpha_{\text{score}} \theta_{t,j}) \left(1 + \frac{\alpha_{\text{unc}}}{n_j + 1}\right)$$
$$\times \exp\left(-\frac{(\theta_{t,i} - \theta_{t,j})^2}{\tau^2}\right),$$

which biases comparisons toward similarly rated models while still favoring strong and under-explored arms.

**Bradley–Terry ranking and convergence.** Given the accumulated pairwise outcomes, LMA fits a Bradley–Terry (BT) model over the $K$ models. In the BT model, each model $k$ has a latent strength parameter $\theta_k$, and the probability that model $i$ is preferred over model $j$ is

$$\Pr(i \succ j) = \sigma(\theta_i - \theta_j) = \frac{1}{1 + \exp(-(\theta_i - \theta_j))}.$$

We estimate $\boldsymbol{\theta}$ via maximum likelihood using iterative gradient ascent, re-fitting the model every 2 rounds (rounds 1, 2, 4, 6, etc.). A model is *feasible* if it satisfies all routing constraints (e.g., price and speed) according to its metadata. The system is declared to have *converged* if the best feasible model passes all constraints for two consecutive rankings (e.g., rankings at rounds 1–2, 2–4, 4–6, or 10–12). This gives an advantage to this baseline since it can converge in 3 iterations minimum and 4 iterations maximum.

C.4.2. COST-CONSTRAINED LMA

We consider a finite pool of $K$ candidate language models, indexed by $i \in \{1, \ldots, K\}$. Each model is associated with static metadata from the model pool, including input and output price estimates. Let

$$c_i^{\text{static}} = c_i^{\text{in}} + c_i^{\text{out}}$$

denote the static per-model cost proxy, where $c_i^{\text{in}}$ and $c_i^{\text{out}}$ are taken from the model-pool table. These static costs are used for cost-sensitive selection, while realized API costs are logged separately during execution. We incorporate parts of HEART-UCB's budget enforcement to LMA, making it cost-constrained.

**Pairwise preference model.** Let $w_{ij}$ denote the number of times model $i$ defeats model $j$, and let

$$N_{ij} = w_{ij} + w_{ji}$$

denote the total number of comparisons between $i$ and $j$. The algorithm fits a Bradley–Terry (BT) model with score vector $\theta \in \mathbb{R}^K$, where

$$\Pr(i \succ j) = \sigma(\theta_i - \theta_j) = \frac{1}{1 + \exp(-(\theta_i - \theta_j))}.$$

A gradient ascent is then performed on the BT log-likelihood. The per-coordinate gradient is

$$g_i(\theta) = \sum_{j \neq i} \left( w_{ij} - N_{ij}\, \sigma(\theta_i - \theta_j) \right),$$

followed by recentering to enforce $\sum_i \theta_i = 0$.

**Cost-sensitive pair selection.** At round $t$, let

$$n_i = \sum_{j=1}^{K} N_{ij} \qquad \text{and} \qquad s_i = \frac{1}{\sqrt{\max(n_i, 1)}}.$$

Thus, $s_i$ is an uncertainty bonus that is larger for under-compared models.

The algorithm forms a cost-adjusted score

$$\tilde{\theta}_i^{(t)} = \theta_i^{(t)} - \beta_2\, c_i^{\text{static}}\, \rho_t,$$

where $\beta_2 \geq 0$ is a cost weight and $\rho_t \geq 0$ is a dynamically updated resource penalty. The anchor distribution is then

$$p_i^{(t)} \propto \exp\!\left(\alpha_{\text{score}}\, \bar{\theta}_i^{(t)}\right) \left(1 + \alpha_{\text{unc}} s_i\right),$$

where

$$\bar{\theta}_i^{(t)} = \tilde{\theta}_i^{(t)} - \frac{1}{K} \sum_{k=1}^{K} \tilde{\theta}_k^{(t)}.$$

With probability $\varepsilon_{\text{rand}}$, the anchor is chosen uniformly at random; otherwise it is sampled from $p^{(t)}$.

Conditioned on anchor $i$, the opponent distribution is

$$q_{j|i}^{(t)} \propto \mathbf{1}[j \neq i]\, \exp\!\left(\alpha_{\text{score}}\, \bar{\theta}_j^{(t)}\right) \left(1 + \alpha_{\text{unc}} s_j\right) \exp\!\left(-\frac{(\tilde{\theta}_i^{(t)} - \tilde{\theta}_j^{(t)})^2}{\tau^2}\right),$$

where $\tau > 0$ biases comparisons toward similarly scored models.

**Observed pairwise outcome.** For a sampled pair $(i, j)$, if the left model is preferred, then $i$ is recorded as the winner; otherwise $j$ is recorded as the winner. The sufficient statistics are updated as

$$w_{ij} \leftarrow w_{ij} + 1 \quad \text{if } i \text{ wins over } j,$$

and

$$N_{ij} \leftarrow N_{ij} + 1, \qquad N_{ji} \leftarrow N_{ji} + 1.$$

**Runtime cost accounting.** Let $r_t^{(L)}$ and $r_t^{(R)}$ denote the realized API costs of the left and right model calls at round $t$. The round cost logged by the implementation is

$$r_t = r_t^{(L)} + r_t^{(R)},$$

and cumulative runtime spend is tracked as

$$S_t = \sum_{u=1}^{t} r_u.$$

The implementation reports budget compliance ex post via

$$S_T \leq B,$$

where $B$ is the user-specified total budget.

**Penalty-mode pacing.** The target per-round spend is defined as

$$c_{\text{target}} = \frac{B}{T + \tau_s},$$

where $T$ is the number of prompts and $\tau_s \geq 0$ is a sensitivity parameter. The resource penalty is updated online as

$$\rho_{t+1} = \max\{0, \ \rho_t + \eta \left( r_t - c_{\text{target}} \right)\},$$

where $\eta > 0$ is a step size. Hence, if realized spend exceeds the target, future expensive models receive a larger penalty through $\tilde{\theta}_i^{(t)}$.

**Best-model selection.** The current best model is selected as:

$$i^{\star} = \arg\max_i \left( \theta_i - \beta_2 \, c_i^{\text{static}} \, \rho_t \right).$$

### C.4.3. RELATIVE UPPER CONFIDENCE BOUND (RUCB)

Our second baseline is **Relative Upper Confidence Bound (RUCB)** (Zoghi et al., 2014). RUCB maintains a pairwise win matrix $W(t) \in \mathbb{N}^{K \times K}$, where $w_{ij}(t)$ is the number of times model $i$ has beaten model $j$ up to round $t$, and $N_{ij}(t) = w_{ij}(t) + w_{ji}(t)$ is the number of comparisons between $i$ and $j$.

**Optimistic confidence bounds and champion set.** For each pair $(i, j)$ we define the empirical win probability

$$\widehat{p}_{ij}(t) \ = \ \begin{cases} \dfrac{w_{ij}(t)}{N_{ij}(t)}, & \text{if } N_{ij}(t) > 0, \\ \frac{1}{2}, & \text{otherwise,} \end{cases}$$

and construct RUCB-style upper confidence bounds

$$u_{ij}(t) \ = \ \widehat{p}_{ij}(t) + \sqrt{\frac{\alpha \log t}{N_{ij}(t)}},$$

with the convention $u_{ij}(t) = 1$ when $N_{ij}(t) = 0$, and $u_{ii}(t) = \frac{1}{2}$ on the diagonal. Intuitively, $u_{ij}(t)$ is an optimistic estimate of the probability that $i$ beats $j$, which shrinks as we collect more comparisons for $(i, j)$. RUCB then forms the *champion set*

$$\mathcal{C}_t \ = \ \left\{ i \in [K] : u_{ij}(t) \geq \tfrac{1}{2} \ \text{ for all } j \in [K] \right\},$$

i.e., arms that are not currently ruled out as Condorcet winners under the optimistic bounds.

We also maintain a *hypothesized best arm* $\mathcal{B}_t \subseteq [K]$, which in our implementation either contains a single index or is empty. If $\mathcal{C}_t = \emptyset$, RUCB chooses a champion uniformly at random from all arms. Otherwise, any hypothesized best arm that is no longer in $\mathcal{C}_t$ is discarded,

$$\mathcal{B}_t \leftarrow \mathcal{B}_t \cap \mathcal{C}_t.$$

If $|\mathcal{C}_t| = 1$, RUCB sets that arm as both the champion and the hypothesized best, $c_t \in \mathcal{C}_t$ and $\mathcal{B}_t = \{c_t\}$. If $|\mathcal{C}_t| > 1$ and $\mathcal{B}_t$ is non-empty, RUCB selects

$$c_t \ = \ \begin{cases} \text{the arm in } \mathcal{B}_t, & \text{with probability } \frac{1}{2}, \\ \text{a uniform arm in } \mathcal{C}_t \setminus \mathcal{B}_t, & \text{with probability } \frac{1}{2}, \end{cases}$$

and if $\mathcal{B}_t$ is empty it samples $c_t$ uniformly from $\mathcal{C}_t$.

**Relative UCB opponent selection.**   Given the champion $c_t$, RUCB treats it as a benchmark and applies a UCB rule over the column of $c_t$. Specifically, it selects an opponent

$$d_t \in \arg \max_{j \in [K] \setminus \{c_t\}} u_{jc_t}(t),$$

breaking ties uniformly at random. The resulting pair $(c_t, d_t)$ is then evaluated on the current HelpSteer2 prompt: both models are invoked via the API (incurring per-round cost equal to the sum of their runtime costs), and the LLM judge compares their metadata against the constraints to declare a winner. If model $i$ is judged preferable to model $j$, we update $w_{ij}(t+1) = w_{ij}(t) + 1$ and leave all other entries of $W$ unchanged.

**Copeland ranking and convergence.**   RUCB is theoretically analyzed in terms of Condorcet regret, but for evaluation we derive a ranking from the empirical win matrix $W(t)$ using normalized Copeland scores. After each round $t$ we compute

$$\widehat{\zeta}_i(t) \;=\; \frac{1}{K-1} \sum_{j \neq i} \mathbb{I}\!\left( \frac{w_{ij}(t)}{w_{ij}(t) + w_{ji}(t) + \mathbb{I}[w_{ij}(t) + w_{ji}(t) = 0]} > \tfrac{1}{2} \right),$$

the fraction of opponents that arm $i$ empirically beats more than half the time. We define the current RUCB-best model as the empirical Copeland winner

$$i_t^\star \in \arg \max_{i \in [K]} \widehat{\zeta}_i(t).$$

As in the other baselines, a model is *feasible* if it satisfies all routing constraints (e.g., price and speed) according to its metadata. We declare that RUCB has *converged* once the RUCB-best model $i_t^\star$ is feasible for 5 consecutive rounds. We report both the final RUCB-best model and the round at the first occurrence of this constraint-met streak, together with the total runtime cost accumulated over all comparisons up until the final round of the streak.

All the runs will be from seed 42 to 46.

### C.4.4. EPSILON GREEDY

We consider the stochastic dueling bandit protocol (Yue & Joachims, 2009) with $K$ models (arms). At each round $t$, the algorithm selects an ordered pair $(c_t, d_t)$ and observes a binary outcome indicating which model is preferred.

**Pairwise win statistics.**   We maintain a win matrix $W(t) \in \mathbb{N}^{K \times K}$, where $w_{ij}(t)$ is the number of times model $i$ has beaten model $j$ up to (and including) round $t$. Define the total number of comparisons between $i$ and $j$ as

$$N_{ij}(t) \;=\; w_{ij}(t) + w_{ji}(t).$$

The empirical win probability estimate is

$$\widehat{p}_{ij}(t) \;=\; \begin{cases} \dfrac{w_{ij}(t)}{N_{ij}(t)}, & \text{if } N_{ij}(t) > 0, \\[2mm] \tfrac{1}{2}, & \text{if } N_{ij}(t) = 0, \end{cases} \qquad \text{with} \qquad \widehat{p}_{ii}(t) = \tfrac{1}{2}.$$

(Thus, unseen pairs are treated as neutral ties in expectation.)

**Mean-score aggregation.**   From the empirical pairwise estimates, we compute a per-model mean score $\mu_i(t)$ as the average probability of beating a uniformly random opponent:

$$\mu_i(t) \;=\; \frac{1}{K-1} \sum_{j \neq i} \widehat{p}_{ij}(t).$$

This is the empirical *Borda score* objective used frequently in dueling/voting bandit variants (Urvoy et al., 2013; Jamieson et al., 2015).

**$\epsilon$-greedy champion selection.**   Given $\mu(t) \in \mathbb{R}^K$, the champion index $c_t$ is drawn by an $\epsilon$-greedy rule (Sutton & Barto, 2018; Auer et al., 2002):

$$c_t \;=\; \begin{cases} \arg \max_{i \in [K]} \mu_i(t), & \text{with prob. } 1 - \epsilon, \\ \text{Uniform}([K]), & \text{with prob. } \epsilon, \end{cases}$$

breaking ties uniformly at random among maximizers.

**Challenger selection.** The challenger is deterministically selected as the best remaining arm under the same mean score:

$$d_t \in \arg \max_{j \in [K] \setminus \{c_t\}} \mu_j(t),$$

again breaking ties uniformly at random.

**Update rule.** After playing the duel $(c_t, d_t)$ we observe the winner $w_t \in \{c_t, d_t\}$. If $w_t = c_t$, we update $w_{c_t d_t}(t+1) = w_{c_t d_t}(t) + 1$; otherwise $w_{d_t c_t}(t+1) = w_{d_t c_t}(t) + 1$. All other entries of $W$ remain unchanged. We then recompute $\widehat{p}_{ij}(t+1)$ and $\mu(t+1)$ from the updated counts.

**Ranking and feasibility.** At any round $t$, we rank models by $\mu_i(t)$ in descending order and define the current best model as

$$i_t^\star \in \arg \max_{i \in [K]} \mu_i(t).$$

As in the other baselines, a model is feasible if it satisfies all routing constraints based on its metadata. We declare convergence once $i_t^\star$ is feasible for 5 consecutive rounds.

## D. Setup

We assume that hidden objectives like helpfulness of an LLM is up to the user's decision in each duel. Instead, we focus on tangible objectives, e.g. input cost, image input feature support, or a large window context.

### D.1. Procedure

We assume a user with a hidden preference that is either unknown or undefined at the beginning of the selection process. Over the course of the interaction, the user gradually refines their choices and ultimately selects the LLM that best aligns with this latent preference.

To simulate a real user, We use prompt engineering techniques such as few-shot prompting (Brown et al., 2020) on LLM-as-the-judge and LLM-as-the-user to increase accuracy. We attempt to mitigate known problems of LLM-as-the-judge such as positional bias or inconsistency (Zheng et al., 2023) by encouraging the LLM to only consider model metadata and user constraints and assess trade offs between them. We also use HelpSteer2 (Wang et al., 2025a) to avoid prompting bias.

We assume a very low cost budget, calculated by the median of cost input divided by the number of models: \$1.25/25 = \$0.05, to observe how the systems react to unrealistic budgets. We also assume a round budget of 10, as most conversations conclude within 10 turns (Deng et al., 2023).

We designed the following setup to assess the limits of CUPID:

- We utilize the routing model at every turn. After each round, the user requests exactly *one* change (e.g. I want a cheaper model).

- We do not perform hyperparameters tuning; instead, we use the configurations from Table 4

- We dropped all columns containing the model name. This disables possible beneficial external knowledge for the routing model.

### D.2. Simulation Environment

D.2.1. USER SIMULATOR

This agent simulates a human user who refines their requirements over time. It receives the metadata of the previously selected model and generates a directional feedback signal (e.g., "I need a cheaper model").

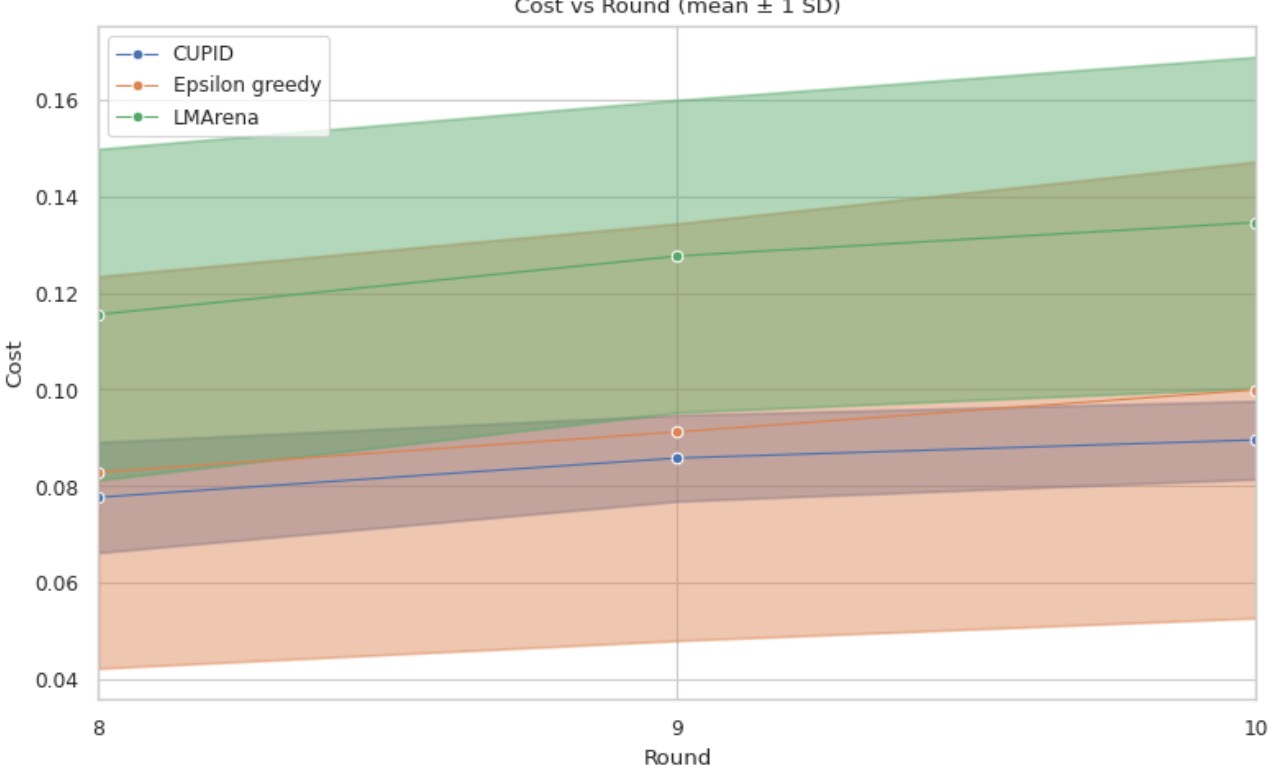

*Figure 7.* Mean cost and standard deviation at final rounds.

---

**System Prompt: User Simulator**

```
You are simulating how a human user refines their preferences over language models.
You will be given:
- A list of formal constraints / objectives they have (possibly empty).
- Metadata for the model that was most recently chosen (or '(none yet)').
Your job:  1) Produce a short natural-language DIRECTION for what the user wants
next.  Examples:  'I want a cheaper model.', 'I need a model with image input.', 'I
want faster responses.', 'I want a model with a newer knowledge cutoff.'.
2) From the list below, focus on at most ONE main change or preference that the
constraint did not meet.  Do not combine multiple items.  [List of Attributes:
intelligence, speed, input_price, etc...]
3) You SHOULD consider the last chosen model's metadata and any explicit constraints.
For example, if the last model is expensive AND slow, the direction might ask for
a cheaper model.  And if the next iteration it is still slow the direction might
ask for a faster model.  Only give comparison style direction, 'cheaper' model, not
cost<=5 model.
4) Output ONLY the DIRECTION sentence, without quotes, without any prefix like
'DIRECTION:'.  No explanations.
5) If all constraints are met, output 'NONE'
```

## D.3. User Types (A1–A4) and Constraint/Budget Construction

**Budget and round limits.** We simulate a strict interaction budget with a maximum number of rounds $T$ and a monetary budget $B$, and use the main paper pacing rule $c_{\text{target}} = B/(T + \tau)$ for the cost controller.

**Constraint Construction.** We sample the $n$ constraints, then for each constraint we sample $k$ values from it. We then get the constraints by setting the minimum or maximum value to the sampled value using the rules in 9

*Table 9.* Constraints Rules.

| Objective | Min/Max | Comparison Type |
|---|---|---|
| intelligence | Max | $\geq$ |
| speed | Max | $\geq$ |
| text-input | Max | $\geq$ |
| image-input | Max | $\geq$ |
| voice-input | Max | $\geq$ |
| video-input | Max | $\geq$ |
| text-output | Max | $\geq$ |
| image-output | Max | $\geq$ |
| audio-output | Max | $\geq$ |
| video-output | Max | $\geq$ |
| reasoning | Max | $\geq$ |
| input-price | Min | $\leq$ |
| cached-input | Max | $\leq$ |
| output-price | Min | $\leq$ |
| window-context | Max | $\geq$ |
| max-output | Max | $\geq$ |
| completion-endpoint | Max | $\geq$ |
| responses-endpoint | Max | $\geq$ |
| assistants-endpoint | Max | $\geq$ |
| batch-endpoint | Max | $\geq$ |
| fine-tuning-endpoint | Max | $\geq$ |
| streaming | Max | $\geq$ |
| function-calling | Max | $\geq$ |
| structured-output | Max | $\geq$ |
| fine-tuning | Max | $\geq$ |
| distillation | Max | $\geq$ |
| predicted-outputs | Max | $\geq$ |
| rate-limit-tier-one | Max | $\geq$ |
| rate-limit-tier-two | Max | $\geq$ |
| rate-limit-tier-three | Max | $\geq$ |
| rate-limit-tier-four | Max | $\geq$ |
| rate-limit-tier-five | Max | $\geq$ |

For example, if algorithm samples the objective "intelligence" and a set of values $\{3, 2, 5, 1\}$ then the constraint will be $\geq 5$.

**User types (A1–A4).** In the OpenAI model-zoo simulations (Table 1), we evaluate four experimental setups (A1–A4) corresponding to four simulated user types that differ in the number of relevant objectives $K \subseteq K'$ (where $K'$ is the full objective set annotated for each model in the zoo). For each setup, $M^\star$ denotes the number of models that are *tied* for best performance under the simulated user's objective set (i.e., multiple models may satisfy the objectives equally well).

**Experiment 1:** $n = 5$, $k = 5$, seed = 42 In this experiment, the user hidden preferences are: a model with output price <= 0.4, tier five rate limit >= 150000000, function calling feature, exists a batch endpoint, and text input feature.

**Experiment 2:** $n = 7$, $k = 5$, seed = 44 In this experiment, the user hidden preferences are: a model with cached input price <= 1.25; supports image input, function calling, text input; does not have image output feature; and exists a batch endpoint and a responses endpoint.

**Experiment 3:** $n = 9$, $k = 5$, seed = 46 In this experiment, the user hidden preferences are: a model with cached input price <= 0.13; supports function calling and text input; does not support video input, image output, and voice input; exists a batch endpoint; and has a max token output >= 100000 and tier three rate limit >= 4000000.

**Experiment 4: Constraints correlation** This experiment allows us to observe whether making a condition weaker accelerate the convergence speed or save more cost. The constraints are the same, but the max token output is reduced to >= 10000 instead of 100000.

| Scale | Ratings (1 = lowest, 5 = highest) |
|---|---|
| **Model Quality (1–5)** | 1 = "Very bad / Far from good"; 
 2 = "Bad model"; 
 3 = "Average model, could be better"; 
 4 = "Model does well for what it should be"; 
 5 = "Aligns very well / Goes beyond expectations". |
| **Budget Compliance (1–5)** | 1 = "Heavily exceeded the budget"; 
 2 = "Somewhat exceeded the budget"; 
 3 = "Just right"; 
 4 = "Saved some money"; 
 5 = "Very efficient cost-wise". |

*Table 10.* Likert anchors shown to participants for final-model evaluation.

# E. Experiment - Human Study

## E.1. User Study Interface

**Interface.** Participants interact with two systems in parallel (CUPID vs. baseline), masked as "System A" and "System B". Each interaction round consists of: (i) a single participant prompt, (ii) two candidate outputs produced *within each system* for that prompt, (iii) the participant selecting the preferred output *within each system* (i.e., two independent within-system comparisons per round), and (iv) the participant can provide optional feedback *for each system*. Figure 8 shows the interaction-phase UI used for entering prompts, viewing side-by-side outputs, and selecting within-system preferences. Figure 9 shows the final evaluation UI.

## E.2. Protocol and Participants

**Study design.** We used an A/B-blinded, within-subject design: each participant experienced both CUPID and a strong adaptive dueling-bandit baseline (**LMA / Arena-style sampling**; see Appendix C.4.1 for implementation details). System identity was masked (A/B), and model identities were hidden throughout the study.

Each session has three stages:

1. **Battle phase:** participants submitted prompts and provided within-system pairwise preferences for each system at each round.

2. **Final model Evaluation phase:** after the interaction phase ended (budget exhausted or rounds completed), each system nominated its final matched model. Participants then evaluated the two final matched models by playing with it and seeing their responses side by side.

3. **User Rating phase:** participants rated each system's final matched model on two 5-point Likert scales (shown in Table 10).

Participants also saw each system's **total accrued cost**, **rounds completed** and the final matched model details before submitting ratings.

**Participants, ethics, and exclusions.** We conducted two distinct user evaluations: a text user study and an image user study. Participants were recruited via convenience sampling through professional and academic networks. Participation was voluntary and uncompensated; the study was open to eligible adults (18+).

Text study sample size was $N_{\text{text}} = 41$ across H1–H3; image study sample size was $N_{\text{img}} = 10$ (H4). Per-paradigm sample sizes were: H1 $N = 14$, H2 $N = 12$, H3 $N = 15$, H4 $N = 10$. The population was predominantly young, with a mean age of 22.32 ($SD = 4.59$). The sample was highly educated: 48.78% held a Bachelor's degree, while 41.46% held a graduate degree. For expert group where major is asked, 100% of participants identified Computer Science or IT as their major. Participants generally reported moderate-to-high AI familiarity.

| Setup | N | Quality rating (Likert 1–5) | | | | Budget compliance (Likert 1–5) | | | |
|---|---|---|---|---|---|---|---|---|---|
| | | W/D/L (%) | Median $\Delta$ | $p$ | $r_{\mathrm{rb}}$ | W/D/L (%) | Median $\Delta$ | $p$ | $r_{\mathrm{rb}}$ |
| H1: constraints (text) | 14 | 35.7/7.1/57.1 | -1.0 | 0.537 | -0.187 | 50.0/42.9/7.1 | 0.5 | 0.031 | 0.833 |
| H2: domain expert (text) | 12 | 16.7/33.3/50.0 | -0.5 | 0.289 | -0.500 | 58.3/25.0/16.7 | 1.0 | 0.180 | 0.556 |
| H3: open preference (text) | 15 | 66.7/20.0/13.3 | 1.0 | 0.058 | 0.590 | 60.0/26.7/13.3 | 1.0 | 0.033 | 0.667 |
| H4: open preference (image) | 10 | 50.0/40.0/10.0 | 0.5 | 0.406 | 0.476 | 70.0/20.0/10.0 | 1.0 | 0.047 | 0.833 |
| Text pooled (H1–H3) | 41 | 41.5/19.5/39.0 | 0.0 | 0.805 | 0.046 | 56.1/31.7/12.2 | 1.0 | 0.001 | 0.692 |
| All pooled (H1–H4) | 51 | 43.1/23.5/33.3 | 0.0 | 0.496 | 0.118 | 58.8/29.4/11.8 | 1.0 | 0.000 | 0.730 |

*Table 11.* **Human-study results by paradigm and pooled cohorts.** For each participant-session and endpoint, let $d_i = r_i^{\mathrm{CUPID}} - r_i^{\mathrm{base}}$ be the paired difference in Likert ratings. **W/D/L (%)** reports the percentage of sessions where $d_i > 0$ (win), $d_i = 0$ (draw), and $d_i < 0$ (loss), respectively. **Median** $\Delta$ is the median of $d_i$. $p$ is from a paired Wilcoxon signed-rank test (two-sided). $r_{\mathrm{rb}}$ is rank-biserial correlation (positive favors CUPID).

## E.3. Outcomes and Statistical Analysis

### E.3.1. ENDPOINTS

Our primary subjective endpoints were the final-model Likert ratings: (i) **Quality rating** and (ii) **Budget compliance rating**, both on 1–5 scales. Our objective logs included: (i) **total accrued API cost** and (ii) **number of interaction rounds used**

### E.3.2. WIN / DRAW / LOSS BREAKDOWN

For each participant-session and each endpoint (Quality or Budget Compliance), we compare CUPID against the baseline using the final-model Likert ratings. Let $d_i = r_i^{\mathrm{CUPID}} - r_i^{\mathrm{base}}$ denote the paired difference for participant-session $i$.

We categorize outcomes as:

- **Win** if $d_i > 0$,

- **Draw** if $d_i = 0$,

- **Loss** if $d_i < 0$.

We report the **percentage** of sessions that fall into each category (Win/Draw/Loss) for each paradigm (H1–H4) and pooled cohorts. The overall percentage is visualized as stacked-bar plot in the main text (e.g., Figure 5(a)), and summarized numerically in Table 11.

### E.3.3. PAIRED TESTS AND EFFECT SIZES

To quantify paired differences beyond the Win/Draw/Loss breakdown, we also analyze the paired Likert differences $d_i$ for each endpoint. Because Likert ratings are ordinal, we use a **paired Wilcoxon signed-rank test** (two-sided) for statistical significance. When ties are prevalent, we additionally report a **paired sign test** as a robustness check.

We report a rank-based effect size, **rank-biserial correlation** $r_{\mathrm{rb}}$ (positive values favor CUPID).

### E.3.4. SUMMARY TABLE

Table 11 summarizes results by paradigm (H1–H4) and pooled cohorts (text-only and all studies).

## E.4. Additional Diagnostics

### E.4.1. CUMULATIVE COST GROWTH OVER INTERACTION ROUNDS

To complement aggregate cost endpoints, we plot the cumulative spend trajectory over rounds for both modalities (Figure 10). For each participant session and system, we compute cumulative cost up to round $t$ by summing all API costs incurred by the system through the end of round $t$.

E.4.2. TOTAL COST VS. SUBJECTIVE BUDGET-COMPLIANCE RATING

To relate subjective budget ratings to objective spending, we plot each session's total cost against the corresponding budget rating (Figure 11).

E.4.3. PER-PARADIGM BREAKDOWN

Table 11 reports results by paradigm (H1–H4) and pooled cohorts. All comparisons are **within-participant (paired)**: each participant evaluated both CUPID and the baseline under the same budget and round constraints. We summarize outcomes with (i) the **Win/Draw/Loss (W/D/L)** percentage breakdown (visualized as stacked bars in the main text) and (ii) paired nonparametric statistics on the Likert differences (median $\Delta$, Wilcoxon $p$, and rank-biserial $r_{\rm rb}$). Because per-paradigm sample sizes are modest (H1: $N = 14$, H2: $N = 12$, H3: $N = 15$, H4: $N = 10$), we interpret per-paradigm results as descriptive and emphasize consistent patterns across cohorts.

**Overall pattern (pooled cohorts).** Across both pooled cohorts, CUPID demonstrates a clear advantage on **budget compliance** but a mixed outcome on **quality**. For budget compliance, CUPID wins substantially more often than it loses (text pooled: 56.1/31.7/12.2; all pooled: 58.8/29.4/11.8 W/D/L), with a positive median improvement (median $\Delta = 1.0$ in both pooled cohorts) and statistically significant paired differences (text pooled: $p = 0.001$, $r_{\rm rb} = 0.692$; all pooled: $p < 0.001$, $r_{\rm rb} = 0.730$). In contrast, pooled **quality** differences are not consistent across participants: the median quality difference is $0.0$ (text pooled and all pooled), with near-zero effect sizes ($r_{\rm rb} = 0.046$ and $0.118$) and non-significant Wilcoxon tests ($p = 0.805$ and $p = 0.496$). Taken together, the pooled results suggest CUPID reliably improves perceived budget efficiency while maintaining, on average, comparable final-model quality.

**H1: constraint-based selection (text).** H1 is the most structured setting: participants were given explicit objectives (2–4 constraints) and model specifications were visible. Here, CUPID tends to favor budget efficiency more strongly than quality: quality outcomes skew toward losses (35.7/7.1/57.1 W/D/L; median $\Delta = -1.0$, $p = 0.537$, $r_{\rm rb} = -0.187$), while budget compliance strongly favors CUPID (50.0/42.9/7.1 W/D/L; median $\Delta = 0.5$, $p = 0.031$, $r_{\rm rb} = 0.833$). This pattern is consistent with a cost-aware strategy that participants perceived as more budget-efficient, but not uniformly better in perceived quality under explicit constraints.

**H2: domain expert selection (text).** H2 evaluates domain-appropriate performance, which is partially structured but often multi-faceted. Quality again trends negative (16.7/33.3/50.0 W/D/L; median $\Delta = -0.5$, $p = 0.289$, $r_{\rm rb} = -0.500$), whereas budget compliance trends positive (58.3/25.0/16.7 W/D/L; median $\Delta = 1.0$, $p = 0.180$, $r_{\rm rb} = 0.556$). While the direction matches the overall budget advantage, the H2 cohort is smaller and the variability in perceived domain quality appears higher, making conclusions less stable.

**H3: open preference selection (text).** H3 is the most open-ended text setting: participants selected based on subjective, latent objectives. In this regime, CUPID tends to improve both outcomes: quality strongly favors CUPID (66.7/20.0/13.3 W/D/L; median $\Delta = 1.0$, $p = 0.058$, $r_{\rm rb} = 0.590$), and budget compliance is significantly improved (60.0/26.7/13.3 W/D/L; median $\Delta = 1.0$, $p = 0.033$, $r_{\rm rb} = 0.667$). Although the quality result is marginal at this sample size, the combined directionality suggests CUPID is particularly effective when the target utility is implicit rather than pre-specified.

**H4: open preference selection (image).** H4 extends open preference selection to text-to-image generation. CUPID shows a strong budget compliance advantage (70.0/20.0/10.0 W/D/L; median $\Delta = 1.0$, $p = 0.047$, $r_{\rm rb} = 0.833$) and a positive but non-significant quality shift (50.0/40.0/10.0 W/D/L; median $\Delta = 0.5$, $p = 0.406$, $r_{\rm rb} = 0.476$). This indicates that the budget-efficiency benefit transfers to the image modality, while quality gains are less decisive at the current sample size.

**Caveat on statistical interpretation.** Unless stated otherwise, $p$-values are reported without correction for multiple comparisons; given small per-paradigm sample sizes, we treat per-paradigm $p$-values as supportive evidence rather than definitive proof and emphasize the pooled-cohort trends and W/D/L distributions.

# F. Experiment - Robustness

## F.1. Model Pool

The details of the model pool that is used for robustness analysis are shown in Table 12.

*Table 12.* Models, Benchmark Performance, and Normalized Scores.

| ID | Model ID | Model | MMLU Pro | AIME2025 | Score |
|----|----------|-------|----------|----------|-------|
| 1 | deepseek/deepseek-v3.2-speciale | DeepSeek V3.2 Speciale | 86.3 | 96.7 | 100 |
| 2 | z-ai/glm-4.7 | GLM-4.7 | 85.6 | 95 | 98.24 |
| 3 | moonshotai/kimi-k2-thinking | Kimi K2 Thinking | 84.8 | 94.7 | 97.93 |
| 4 | deepseek/deepseek-v3.2 | DeepSeek V3.2 | 86.2 | 92 | 95.14 |
| 5 | x-ai/grok-4-fast | Grok 4 Fast | 85 | 89.7 | 92.76 |
| 6 | x-ai/grok-4.1-fast | Grok 4.1 Fast | 85.4 | 89.3 | 92.35 |
| 7 | z-ai/glm-4.6 | GLM-4.6 | 82.9 | 86 | 88.93 |
| 8 | z-ai/glm-4.6v | GLM-4.6V | 79.9 | 85.3 | 88.21 |
| 9 | anthropic/claude-haiku-4.5 | Claude 4.5 Haiku | 76 | 83.7 | 86.56 |
| 10 | minimax/minimax-m2.1 | MiniMax-M2.1 | 87.5 | 82.7 | 85.52 |
| 11 | z-ai/glm-4.5-air | GLM-4.5-Air | 81.5 | 80.7 | 83.45 |
| 12 | minimax/minimax-m2 | MiniMax-M2 | 82 | 78.3 | 80.97 |
| 13 | z-ai/glm-4.5 | GLM-4.5 | 83.5 | 73.7 | 76.22 |
| 14 | z-ai/glm-4.5v | GLM-4.5V | 78.8 | 73 | 75.49 |
| 15 | google/gemini-2.5-flash | Gemini 2.5 Flash | 83.2 | 60.3 | 62.36 |
| 16 | moonshotai/kimi-k2-0905 | Kimi K2 0905 | 81.9 | 57.3 | 59.26 |
| 17 | google/gemini-2.5-flash-lite | Gemini 2.5 Flash-Lite | 75.9 | 53.3 | 55.12 |
| 18 | openai/gpt-5-chat | GPT-5 (ChatGPT) | 82 | 48.3 | 49.95 |
| 19 | openai/gpt-4.1-mini | GPT-4.1 mini | 78.1 | 46.3 | 47.88 |
| 20 | mistralai/mistral-large-2512 | Mistral Large 3 | 80.7 | 38 | 39.3 |
| 21 | mistralai/devstral-2512 | Devstral 2 | 76.2 | 36.7 | 37.95 |
| 22 | openai/gpt-4.1 | GPT-4.1 | 80.6 | 34.7 | 35.88 |
| 23 | mistralai/ministral-8b-2512 | Ministral 3 8B | 64.2 | 31.7 | 32.78 |
| 24 | mistralai/ministral-14b-2512 | Ministral 3 14B | 69.3 | 30 | 31.02 |
| 25 | openai/chatgpt-4o-latest | GPT-4o | 80.3 | 25.7 | 26.58 |
| 26 | openai/gpt-4.1-nano | GPT-4.1 nano | 65.7 | 24 | 24.82 |
| 27 | google/gemma-3-27b-it | Gemma 3 27B Instruct | 66.9 | 20.7 | 21.41 |
| 28 | meta-llama/llama-4-maverick | Llama 4 Maverick | 80.9 | 19.3 | 19.96 |
| 29 | google/gemma-3-12b-it | Gemma 3 12B | 59.5 | 18.3 | 18.92 |
| 30 | openai/gpt-4o-mini | GPT-4o mini | 64.8 | 14.7 | 15.2 |
| 31 | meta-llama/llama-4-scout | Llama 4 Scout | 75.2 | 14 | 14.48 |
| 32 | google/gemma-3-4b-it | Gemma 3 4B | 41.7 | 12.7 | 13.13 |
| 33 | meta-llama/llama-3.3-70b-instruct | Llama 3.3 70B Instruct | 71.3 | 7.7 | 7.96 |
| 34 | meta-llama/llama-3.1-8b-instruct | Llama 3.1 8B Instruct | 47.6 | 4.3 | 4.45 |
| 35 | meta-llama/llama-3.1-70b-instruct | Llama 3.1 70B Instruct | 67.6 | 4 | 4.14 |
| 36 | meta-llama/llama-3.2-3b-instruct | Llama 3.2 3B Instruct | 34.7 | 3.3 | 3.41 |
| 37 | meta-llama/llama-3.2-1b-instruct | Llama 3.2 1B Instruct | 20 | 0 | 0 |

## F.2. Setup

Routing Model Prompt: Appendix B.1.2

Scorer: A wins if $\text{AIME}_A > \text{AIME}_B$

Dataset: Table 12 will be shuffled using the seed of current run

Seed: 42 to 44 (3 runs)

Hyperparameter: Table 4

**CUPID routing prompt:**

- CUPID with noisy feedback: The context consists of both the AIME and MMLU-Pro benchmarks.

  - Adversarial Feedback: "Give me a model that is bad at math"
  - Static Feedback: "CUPID in the Model Zoo: Online Matchmaking for Selecting Your Dream LLM"
  - Meaningless Feedback: "abcdefghijklmnopqrstuvwxyz."

- CUPID with murky objectives: The context consists only of MMLU-Pro benchmark–"Give me a good math model"

- CUPID: The context consists of both the AIME and MMLU-Pro benchmarks–"Give me a good math model"

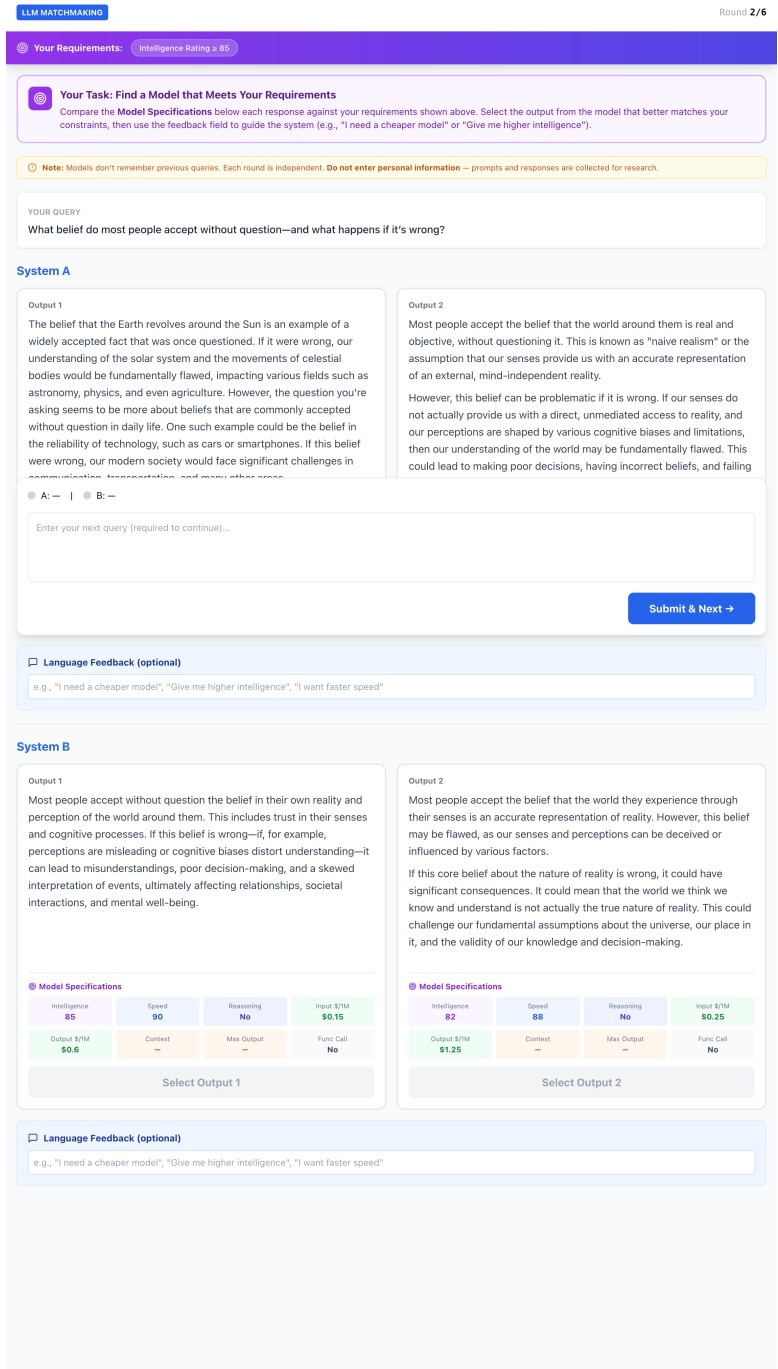

*Figure 8.* User study interface for the interaction phase (within-system duels).

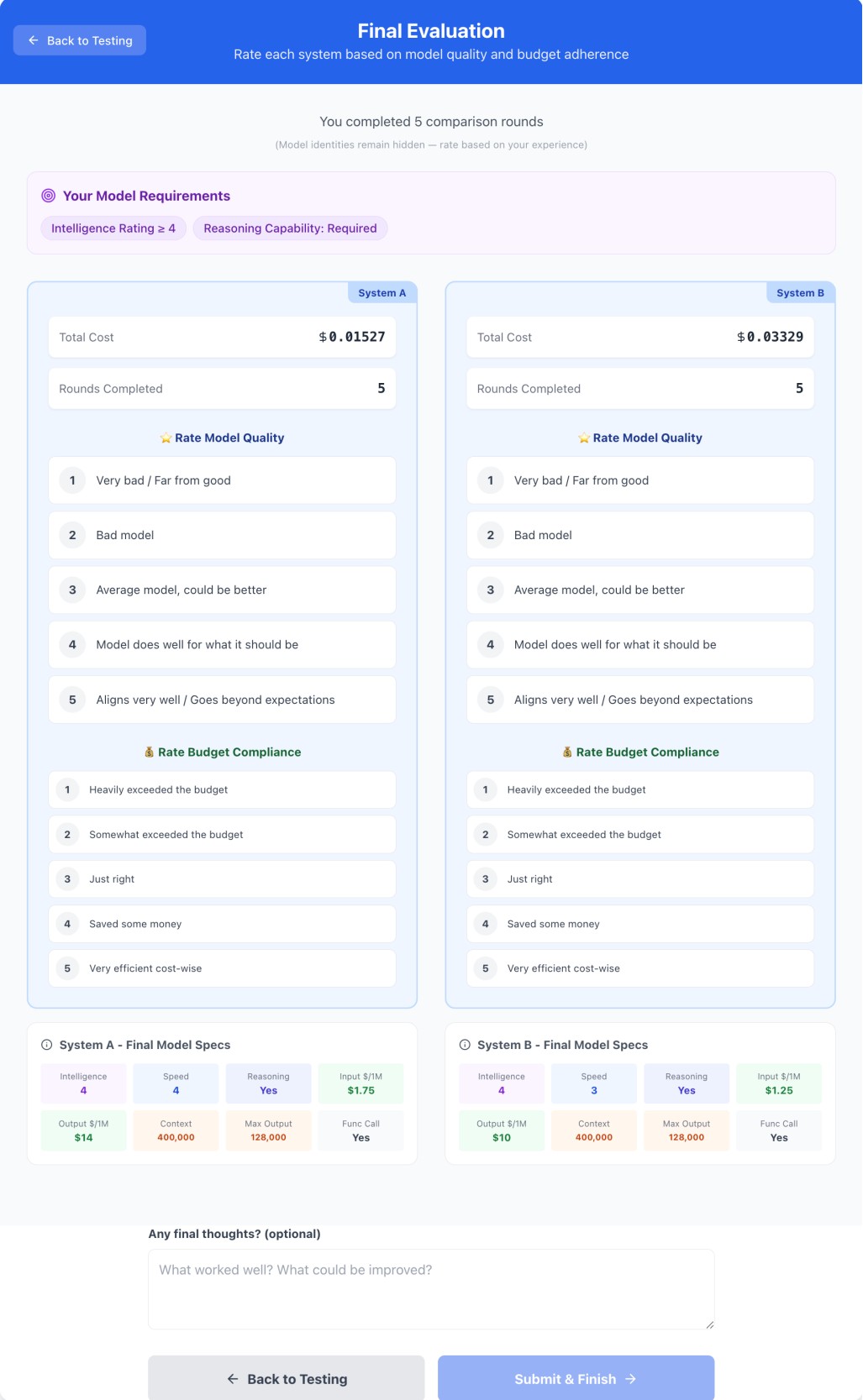

*Figure 9.* Final-model evaluation interface. Participants rate each system's final matched model.

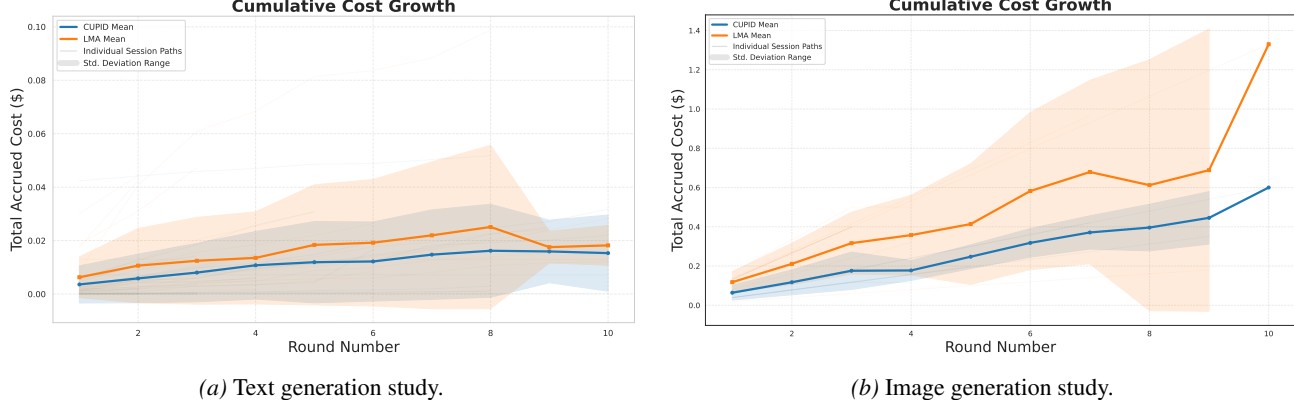

*(a)* Text generation study.

*(b)* Image generation study.

*Figure 10.* **Cumulative cost growth during the interaction phase.** The y-axis reports cumulative spend. The x-axis is interaction round. This diagnostic provides a time-resolved view of budget consumption, complementing aggregate cost outcomes.

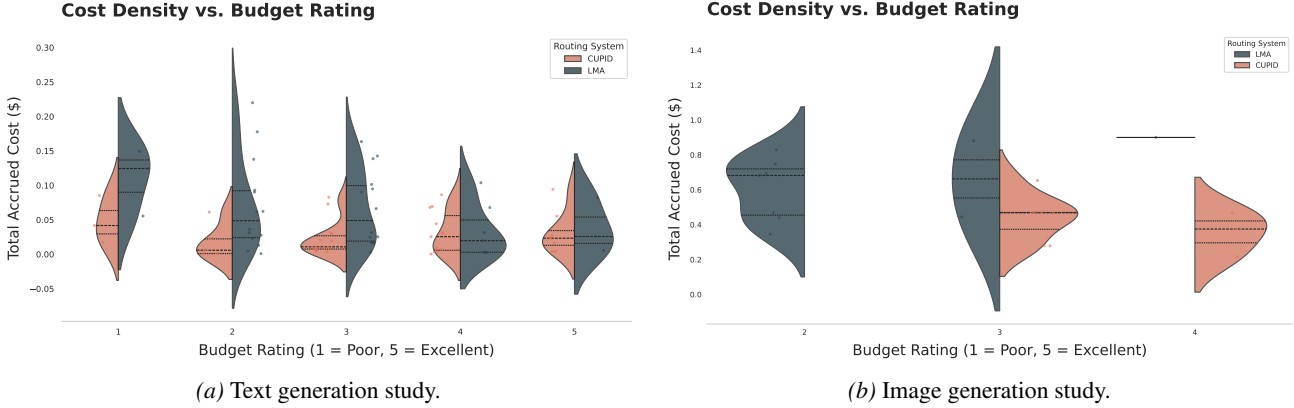

*(a)* Text generation study.

*(b)* Image generation study.

*Figure 11.* **Objective spending vs. subjective budget rating.** Each point corresponds to one participant session. The x-axis is the budget rating (Likert 1-5), and the y-axis is the participant's budget-compliance rating for the corresponding system. This figure links the subjective budget endpoint to an objective measure of budget usage.

