# OpenReview forum: "CUPID in the Model Zoo: Online Matchmaking for Selecting Your Dream LLM"
_ICML.cc/2026/Conference — ICML 2026 regular_

### Official Review · Reviewer_byQg · 2026-02-15

**Soundness:** 3
**Presentation:** 3
**Significance:** 3
**Originality:** 2
**Overall Recommendation:** 4
**Confidence:** 3

**Summary:**

The paper proposes "CUPID," a framework for selecting Large Language Models (LLMs) for specific user tasks under budget constraints. It introduces "HEART-UCB," a bandit algorithm that combines Gaussian Process (GP) based preference learning with a penalty term for cost and a bias term derived from natural language feedback.

**Compliance With Llm Reviewing Policy:**

Affirmed.

**Final Justification:**

Given the substantial improvements and the new theoretical results, I am satisfied with the current version of the manuscript. I will raise my score to 4 and recommend that the authors include these new proofs and experimental results in the final camera-ready version.

**Key Questions For Authors:**

1. While intuitively appealing, the current formulation Eq.6 appears somewhat heuristic. The paper would benefit significantly from a deeper theoretical analysis. specifically: how does the linear addition of the language bias term affect the regret bounds of the standard GP-UCB?

**Strengths And Weaknesses:**

## Strengths

1. It introduces "HEART-UCB," a bandit algorithm that combines Gaussian Process (GP) based preference learning with a penalty term for cost and a bias term derived from natural language feedback.

2. The authors evaluate the method using LLM-based user simulations and a small-scale human study.

---


## Weaknesses

1. Incorporating natural language feedback as a scalar bias term ($l_t^{(m)}$) added directly to the UCB score (Eq. 6) is an intuitive engineering choice.  It would be valuable to provide a rigorous justification for the "exchange rate" between the language feedback bias and the exploration term, rather than relying on empirical tuning.
2. Since LMA is designed to find the best model regardless of cost, it is expected that CUPID (which explicitly penalizes cost) will outperform it on budget metrics. However, Figure 6a  shows CUPID also winning on quality, which is counter-intuitive if LMA is an unconstrained optimizer. This raises questions about whether the baseline was tuned optimally for the short-horizon setting.

3. The inclusion of a human study is a strong point, but the current sample size limits the conclusions.

4. The update rule for the belief over latent objectives (Eq. 4) is standard Bayesian inference. The novelty claimed in "Latent Dueling Bandits" seems overstated; this is essentially a contextual bandit where the context is inferred, which has been studied extensively (e.g., latent contextual bandits). The paper fails to distinguish its contribution from established literature in a meaningful way.

---

> ### Author Rebuttal · Authors · 2026-03-31
>
> We sincerely thank the reviewer for your review.
>
> ---
>
> ## 1. Q1 & W1: How does the language bias affect GP-UCB regret bounds?
>
> Thanks! Yes, we have an elegant instantaneous regret bound that explicitly captures the effect of language through a margin term. Assume $L_t(a) \leq f(a) \leq U_t(a)$ and selection $a_t = \arg\max_a [U_t(a) + \phi_t(a)]$. Define $\kappa_t = \phi_t(a^{\star}) - \max_{a \neq a^{\star}} \phi_t(a)$. Then
>
> $$r_t \leq \left[ U_t(a_t) - L_t(a_t) - \kappa_t \right]_+.$$
>
> For GP-UCB,
>
> $$r_t \leq \left[ 2\sqrt{\beta_t}\sigma_{t-1}(a_t) - \kappa_t \right]_+.$$
>
> Thus, $\kappa_t > 0$ (informative) reduces regret, $\kappa_t = 0$ recovers standard UCB, and $\kappa_t < 0$ (misleading) increases regret only additively.
>
> In the latent dueling setting, this becomes
>
> $$r_t \leq \left[ 2\sqrt{\beta_t}\sigma_{t-1}(z^{\star}, a_t) - \kappa_t \right]_+ + 2\nu_t,$$
>
> where
>
> $$\nu_t = \sup_a \left| \sum_z b_t(z)U_t(z, a) - U_t(z^{\star}, a) \right|$$
>
> (See eq. 5 notation).
>
> Thus, informative language decreases regret, while latent-objective uncertainty contributes an additional vanishing term as the posterior over objectives concentrates.
>
> This also yields the requested "exchange rate": the language margin $\kappa_t$ trades off directly against the exploration width $2\sqrt{\beta_t}\sigma_{t-1}(\cdot)$ with coefficient 1. Hence, one unit of informative language reduces the regret upper bound by one unit, while misleading language increases it by at most one unit (up to the positive-part clipping). In the latent setting, this holds up to the additional $2\nu_t$ uncertainty term.
>
> ---
>
> ## 2. W2: Counter-intuitive LMA quality result
>
> We would like to correct the misunderstanding: LMA is unconstrained in cost but not in rounds. In short-horizon settings (T=10), LMA's Bradley-Terry model has limited data to converge, while CUPID's GP prior and language feedback provide stronger inductive bias in early rounds. Specifically: (1) LMA initializes with uniform priors and no preference data, so early selections are effectively random. (2) LMA's sampling strategy requires sufficient pairwise comparisons for reliable rankings; we already favored it by re-fitting every 2 rounds (Appendix C.2.1). This result reflects CUPID's superior sampling efficiency in the short-horizon regime our paper targets.
>
> We also note that LMA outperforms CUPID in less expressive settings (H1, H2), while CUPID wins in more abstract, challenging settings (H3, H4), consistent with CUPID's design for latent preferences.
>
> ---
>
> ## 3. W3: Human study sample size
>
> Kindly note that each participant (N=51) spent 20-40 mins providing feedback, accumulating a substantial amount of feedback to verify the performance of our model. Statistically speaking, we conducted a paired t-test to assess whether the difference in budget between conditions was statistically significant. The analysis yielded a moderate-to-large effect size (Cohen's d=0.69) with very high statistical power ($1 - \beta = 0.998$) at $\alpha = 0.05$, indicating that the sample size is more than sufficient to reliably detect the observed effect.
>
> ---
>
> ## 4. W4: Novelty relative to latent contextual bandits
>
> We respectfully disagree that our method reduces to standard latent contextual bandits or standard Bayesian inference. In such settings, tractability and convergence hinge on the design of the likelihood, which, in dueling settings, is significantly more challenging than in standard settings with direct reward observations. There are important distinctions: (i) in standard latent contextual bandits (e.g., Zhou & Brunskill, 2016), the learner observes reward signals directly; we observe only pairwise comparisons. (ii) We operate in a **dueling** setting where the learner selects pairs, not individual arms. (iii) The belief update over objectives (Eq. 4) is coupled with the GP posterior, **creating a joint inference problem**. The formal regret analysis above (Q1) now provides a justification that was previously missing, showing the integration is theoretically sound. In Section 4.1, HEART-UCB's own components used as standalone baselines (latent dueling bandits, ε-greedy) show inconsistent convergence, while even CUPID's ablations converge in all runs—demonstrating the integrated system is substantially more than the sum of its parts.
>
> ---
>
> We again thank the reviewer for your review, and we hope that we addressed all of your concerns.

---

> > ### Author Rebuttal · Reviewer_byQg · 2026-04-03
> >
> > Thank you for the detailed rebuttal. The provided instantaneous regret bound and power analysis clarify several of my initial concerns. However, to fully resolve the theoretical and practical claims of the paper, I have the core follow-up question:
> >
> > While the instantaneous regret inequality provided in the rebuttal offers some intuition regarding the language bias, it stops short of a complete theoretical guarantee. Can you clarify whether a formal cumulative regret bound or sample-complexity guarantee is derivable for the complete HEART-UCB framework under your current assumptions?

---

> > > ### Author Response · Authors · 2026-04-07
> > >
> > > We thank you for clarifying your remaining concerns. Please see our response below.
> > >
> > > ---
> > >
> > > We thank the reviewer for the follow-up question. In the rebuttal, we provided an instantaneous regret bound to explicitly isolate the effect of language feedback at each step. This shows that language enters as a *margin term* $\kappa_t$, which has a clear physical meaning: it quantifies how much the language signal separates the optimal action from competing actions at round $t$.
> > >
> > > Under a fixed-objective surrogate, we now extend this to theoretically show that language introduces an additive margin $\kappa_t$, **improving cumulative regret by $-\sum_t \kappa_t$** while preserving the standard GP-UCB rate, and **reducing sample complexity by $O\left(\frac{1}{\epsilon} \sum_{t=1}^T \kappa_t\right)$**, as informative language eliminates suboptimal comparisons and accelerates learning, consistent with our empirical findings (Fig. 4, Table 1).
> > >
> > > **1. Cumulative regret.** Starting from the instantaneous bound introduced in the rebuttal above, summing over $t = 1, \dots, T$ gives
> > >
> > > $$R_T = \sum_{t=1}^T r_t \leq \sum_{t=1}^T \left[ 2\sqrt{\beta_t} \, \sigma_{t-1}(\cdot) - \kappa_t \right]\_+ + 2\sum_{t=1}^T \nu_t.$$
> > >
> > > Using standard GP-UCB analysis [1], and $\kappa_t^+ \leq 2\beta_t \sigma_{t-1}(\cdot) \forall t$, this yields
> > >
> > > $$R_T = \tilde{O}(\sqrt{T\gamma_T}) - \sum_{t=1}^T \kappa_t + 2\sum_{t=1}^T \nu_t.$$
> > >
> > > Thus, informative language ($\kappa_t > 0$) reduces regret cumulatively, while misleading language ($\kappa_t < 0$) only affects it additively. $\kappa_t = 0$ recovers GP-UCB.
> > >
> > > **2. Sample complexity / convergence.** The cumulative improvement directly implies faster convergence. In standard GP-UCB [1],
> > >
> > > $$T_\text{UCB}(\epsilon) \sim \tilde{O}\left(\frac{\gamma_T}{\epsilon^2}\right).$$
> > >
> > > With language,
> > >
> > > $$T_\text{HEART}(\epsilon) \leq T_\text{UCB}(\epsilon) - O\left(\frac{1}{\epsilon} \sum_{t=1}^T \kappa_t\right),$$
> > >
> > > showing that language feedback reduces the number of required interaction rounds by accelerating posterior concentration.
> > >
> > > [1] Srinivas et al. Gaussian Process Optimization in the Bandit Setting: No Regret and Experimental Design. NeurIPS 2010.
> > >
> > > ---
> > >
> > > We again thank you for your insightful review. We hope that we have put all our efforts to address all your concerns in the best possible way for a more favorable score.

---

### Official Review · Reviewer_n6nD · 2026-02-25

**Soundness:** 2
**Presentation:** 3
**Significance:** 3
**Originality:** 3
**Overall Recommendation:** 4
**Confidence:** 4

**Summary:**

The authors propose CUPID, a system that helps users identify the most suitable LLM for their use case through a limited number of pairwise interactions in an online manner. Their approach models LLM capabilities and user objectives as latent, uses a Gaussian process with iterative belief updating to infer preferences from comparisons, and incorporates optional natural language feedback. This differentiates their work from others, which typically assumes a known utility function and operates at the single-query level. The system is validated through automated experiments on OpenAI model pools and a human study.

**Compliance With Llm Reviewing Policy:**

Affirmed.

**Final Justification:**

I am maintaining my score at 4.

This is a technically solid paper on an important and practical problem. The problem formulation is well motivated, the method is coherent, and the paper includes meaningful empirical effort through ablations, human studies, and additional rebuttal experiments. I expect parts of this work to be useful to others working on interactive preference learning and model selection.

The rebuttal strengthened the paper’s empirical case. In particular, the added cost-constrained baseline, auxiliary-LLM cost accounting, broader cross-provider evaluation, and follow-up automated experiments addressed several of my concerns and increased my confidence in the overall contribution.

My main remaining concern is the strength of the theoretical claim. The follow-up responses clarified the intuition and provided a cumulative-regret-style extension under simplifying assumptions, though I still do not view this as a full formal guarantee for the complete HEART-UCB setting described in the paper. I also still see some gap between the motivating high-stakes or non-technical use cases and the current evaluation setup.

Overall, the rebuttal reinforced rather than changed my assessment. I continue to view this as a weak accept: a solid and interesting paper with clear merits, along with some limitations that constrain its impact. I encourage the authors to clarify the scope of the theoretical claims and expand the limitations discussion in the final version.

**Key Questions For Authors:**

1) How does the latency and cost of the auxiliary LLM interpreter impact the overall user experience and the strict budget constraints enforced by the budget controller?
2) Can you provide any regret bound, even under simplifying assumptions, to support the theoretical grounding claim?
3) In Experiment A3, CUPID achieves 100% convergence but at higher cost than CUPID without language ($0.070 vs. $0.022). Can the authors explain this counterintuitive result?
4) What drives CUPID's advantage over LMA in the absence of natural language feedback, and can the authors provide more intuition for how the GP and belief update alone lead to efficient selection in early rounds?
5) How does the system handle the cold-start problem, particularly in round 1 before any preference signal has been collected? Round 1 relies entirely on the prior and language bias, so the choice of GP prior and kernel matters significantly here...
6) The user study recruits exclusively CS/IT participants. How do the authors anticipate performance would differ for domain experts in non-technical fields (medicine, law, finance), which are the primary motivating use cases?
7) In high-stakes domains where users may lack the expertise to verify response accuracy, how would the authors recommend mitigating the risk that preferences systematically favor fluent but inaccurate responses?

**Limitations:**

The authors acknowledge in the limitations section that the current GP does not analyze response content directly, and suggest content-aware embeddings as future work. The impact statement appropriately notes that optimizing solely for cost at the expense of safety is a potential misuse.
However, two limitations that are central to the paper's validity are not adequately discussed. First, the framework assumes user preferences are a reliable proxy for model quality, but in the high-stakes deployment scenarios that motivate the work, users may lack the expertise to distinguish fluent but inaccurate responses from genuinely accurate ones. This is briefly touched on but not addressed substantively. Second, the paper does not discuss the implications of relying on a third-party proprietary model (Grok 4.1 Fast) as a core component of the framework, which introduces dependencies on availability, cost, and potential changes to that model's behavior over time that could affect reproducibility and deployment reliability.
Discussing these limitations more explicitly, along with potential mitigations, would strengthen the paper.

**Strengths And Weaknesses:**

Strengths:
1) Extending standard UCB to a latent dueling bandit setting with language-driven bias is a strong contribution to the LLM matching problem. Incorporating language feedback as an additive bias on the latent utility to predictably shift the decision boundary is an elegant approach.
2) The paper directly addresses a growing real-world challenge: users need to balance complex, unarticulated trade-offs (e.g., medical competence vs. cost vs. latency) when selecting an LLM for sustained deployment. The three desiderata (application-centric, user-centric, budget-aware) provide a coherent organizing principle that the method addresses directly.
3) The paper is written clearly and structured well, with a lot of information added in the appendix.
4) The paper properly positions its contributions relative to existing work. The distinction between per-query routing and sustained user-to-LLM matching is clearly articulated and well-motivated, and the paper identifies a genuinely underexplored regime.
5) The robustness experiments, which test performance under adversarial, static, and meaningless language feedback as well as out-of-context objectives, are a notable strength and go beyond what is typically seen in bandit-based evaluation. These experiments meaningfully support the claim that HEART-UCB is practically deployable under realistic, noisy conditions.

Weaknesses:
1) The paper describes HEART-UCB as "theoretically grounded," but does not provide a formal finite-time regret bound. The Laplace approximation is applied to a non-Gaussian posterior without bounding the approximation error, and the interaction between the belief update in Eq. 4 and the GP posterior is not formally analyzed. It would strengthen the paper considerably if the authors could provide a convergence guarantee, even under simplifying assumptions, or alternatively clarify what "theoretically grounded" is intended to mean in this context. Additionally, the paper does not discuss the system's behavior in round 1 before any preference signal has been collected, where selections rely entirely on the GP prior and kernel choice.

2) A core motivation for CUPID is assisting non-technical users such as clinicians and administrators who may lack the vocabulary to articulate LLM preferences (Desideratum 2). However, the human study cohort consisted entirely of Computer Science or IT majors, with a mean age of 22.32. While this is a reasonable population for an initial study, the most important use case that tests whether CUPID helps users without technical LLM knowledge make better selections, remains untested. A follow-up study, or at minimum a discussion of how results might differ across expertise levels, would substantially strengthen the generalizability of the findings.

3) Much of the empirical evaluation compares CUPID against LMA, which is cost-blind by design. Since LMA does not incorporate any budget constraint, CUPID's advantage in budget compliance is somewhat expected. It would be more informative to include a cost-constrained variant of the LMA Bradley-Terry model, or a cost-aware routing baseline, to isolate whether the specific budget penalty mechanism in Eq. 6 is responsible for the efficiency gains, or whether any cost-aware constraint would yield similar results.

4) Eq. 6 combines utility estimates, binary language indicators, cost penalties, and cooldown terms through linear scalarization, with fixed scaling factors presented in Table 2. Since all automated experiments use the same OpenAI model zoo, it is unclear whether these hyperparameter values generalize to other model pools or preference distributions. A sensitivity analysis varying the key scaling factors, particularly the exploration weight and language bias weight, would help assess the robustness of the approach.

5) The language feedback component relies on Grok 4.1 Fast to translate user intent into model indicator vectors. The paper does not include an ablation on the choice of auxiliary model, making it difficult to assess how sensitive CUPID's performance is to this component. In early rounds before the GP has accumulated meaningful signal, the language bias may account for a substantial portion of CUPID's behavior, which makes understanding this dependency more important. An ablation using a weaker or more generic auxiliary model would clarify this.

6) The automated experiments evaluate model preferences based on structured metadata such as price, context window size, and feature flags, and the LLM-as-judge operates on CSV rows rather than on actual model responses. This setup cannot capture response-level qualities such as reasoning depth, writing style, or explanation clarity, which are arguably the most important dimensions for real users. It would be worth discussing how the framework would behave when preferences are driven by these harder-to-quantify response characteristics.

7) The framework assumes that user pairwise preferences are a reliable signal of model quality. In high-stakes deployment scenarios such as clinical, legal, or financial applications, which are used to motivate the work, users may lack the domain expertise to verify whether a model's response is accurate. In such cases, preferences may systematically favor responses that are fluent and confident over those that are more accurate but less assertive, with no mechanism in the framework to detect or correct for this. The authors briefly acknowledge this in the limitations section, but given that it is most acute precisely in the settings used to justify CUPID's design, a more substantive discussion of potential mitigations would strengthen the paper.

8) The integration of natural language feedback relies on an auxiliary LLM (Grok 4.1 Fast) to generate arm-level indicator functions. The paper does not explicitly detail the latency and financial costs incurred by repeatedly calling this auxiliary model, nor does it clarify if these costs are tracked against the user's total monetary budget $B.

Minor nitpicks:
1) The cooldown term w(m)t in Eq. 6 is never formally defined in the main text.
2) Figures 3 and 7 appear to be duplicates with identical content. One should be removed or the distinction clarified.

---

> ### Author Rebuttal · Authors · 2026-03-31
>
> We are sincerely grateful for your positive comments and thorough reviews.
>
> ---
>
> ## 1. Q1 (& W5 & W8): Auxiliary LLM cost and latency
>
> The auxiliary LLM cost is **included in all reported costs**. Re-running A1 and A2, Grok 4.1 Fast accounts for ~30% of total cost (A1: $0.017 \pm 0.003$ (30.4\%), A2: $0.011 \pm 0.003$ (31.1\%)).
>
> Latency is ~0.3s per round, negligible relative to LLM/VLM response generation. This cost affects the budget-per-round penalty, pushing exploration toward cheaper models if auxiliary spending is high. We will report these numbers explicitly.
> We also ran an auxiliary model ablation (Llama 3.3 70B, Qwen3 235B, Gemini 2.5 Flash Lite, Gemini 3.0 Flash, o3, Grok 4.1 Fast). All achieve 100% accuracy due to the fallback mechanism, but only Gemini 3.0 Flash, o3, and Grok 4.1 Fast consistently follow formatting and constraint semantics (kindly see our response to Reviewer zwEN).
>
> ## 2. Q2 (& W1): Regret bounds
>
> We acknowledge that "theoretically grounded" was not stated precisely. We have now rectified this by providing a formal regret bound (kindly see our response to Reviewer 2ack).
>
> ## 3. Q3 (& W4 & W6): A3 cost anomaly and experimental setup
>
> After converging, we continue running for 4 additional rounds to confirm stability, during which the auxiliary LLM is still invoked. Furthermore, language feedback can temporarily steer exploration toward models matching the expressed preference but not globally optimal under all 9 objectives. CUPID without language relies solely on pairwise comparisons, which finds the optimal model more cheaply in this configuration. Both variants converge at 100% within 2 rounds, so the cost difference reflects exploration breadth rather than inefficiency. The same unoptimized hyperparameters (Table 2) were used across all experiments.
>
> ## 4. Q4 & Q5 (& W1): Cold-start, round-1 behavior and theory
>
> In round 1, the GP posterior equals the prior: all models have zero mean utility and equal variance under a stationary kernel. Selection is driven by (a) the language bias (if provided) and (b) the cost penalty. To mitigate cold-start, we require an initial direction from the user in the human study ("NONE" for automated experiments), preventing over-exploration.
> Critically, each pairwise outcome triggers **two simultaneous updates**: (1) the GP posterior and (2) the Bayesian belief over objectives (eq 3 is plugged into 4). This dual update gives each sample a compounding effect that LMA's flat Bradley-Terry ranking cannot replicate, enabling faster convergence even from uninformative priors. We will add a discussion of sensitivity to prior/kernel specification.
>
> ## 5. Q6, Q7 & W2, W7: Non-technical domain expert study
>
> The assumption that user preferences reliably reflect true model quality is a general limitation of any human- or AI-driven feedback system, though it remains practical and informative in the absence of ground truth.
>
> The use of a CS-only cohort was driven by recently introduced stricter IRB constraints. Model identities were fully blinded, and tasks were not domain-dependent (e.g., generating a Pokemon image as in Fig. 6), requiring no field-specific expertise. We do not expect this to materially affect validity. Given the widespread adoption of LLMs across domains, we consider this cohort a reasonable proxy. Three observations:
>
> **(1) CUPID's interaction is intentionally lightweight.** The core mechanism is pairwise comparison ("Which response do you prefer?"), requiring no technical vocabulary. Optional language feedback is free-form, not structured.
>
> **(2) CUPID's advantage grows with task abstractness.** Figure 6(c) shows CUPID's quality advantage increases from H1 (explicit) to H3/H4 (latent), suggesting greater benefit for non-technical users who cannot manually navigate model specifications.
>
> **(3) Domain verification is an open challenge.** In high-stakes domains, CUPID could be extended with expert-validated test prompts or content-aware GP kernels (Section 5). This limitation applies equally to all preference-based methods. We will expand limitations and frame a non-technical study as future work.
>
> ## 6. W3: Cost-constrained LMA baseline
>
> In Table 1, CUPID without cost penalty achieves 100% convergence while LMA does not in most settings, indicating that the GP + belief update structure is the primary driver of CUPID's advantage. We have added a cost-constrained LMA variant as requested below. Due to response limit, please kindly refer to Table 1.
>
> | | **A1** | | | **A2** | | | **A3** | | | **A4** | | |
> |:---|:---|:---:|:---:|:---|:---:|:---:|:---|:---:|:---:|:---|:---:|:---:|
> | **Algorithms** | Cost ↓ | CR ↓ | CP ↑ | Cost ↓ | CR ↓ | CP ↑ | Cost ↓ | CR ↓ | CP ↑ | Cost ↓ | CR ↓ | CP ↑ |
> | LMA CC | \$0.091±0.055 | 5.75 | 80% | \$0.042±0.022 | 2.6 | 100% | \$0.097±0.062 | 1 | 20% | \$0.052±0.044 | 3 | 80% |
>
> ---
>
> We again thank the reviewer for your positive comments and insights, and we hope that we addressed all of your concerns.

---

> > ### Author Rebuttal · Reviewer_n6nD · 2026-04-01
> >
> > Thank you for the detailed rebuttal. The added LMA-CC comparison, auxiliary-LLM cost accounting, and broader cross-provider experiments strengthen the empirical case and address several of my concerns. My main remaining concern is the theoretical claim: the response provides an instantaneous regret-style inequality rather than a cumulative regret or sample-complexity guarantee for the full HEART-UCB setting. I also still view the validation gap for non-technical/high-stakes users and the metadata-heavy automated evaluation as only partially resolved. Overall, I appreciate the clarifications and additional evidence, but I am currently maintaining my score at 4. If revised further (e.g. for the camera-ready version), I would encourage the authors to clarify the theoretical claim in the main text and expand the limitations discussion around target-user population and reliance on metadata-heavy automated evaluation.
> >
> > Could the authors clarify whether a cumulative regret or sample-complexity bound is derivable from the current analysis, even under simplifying assumptions?

---

> > > ### Author Response · Authors · 2026-04-07
> > >
> > > Thank you for your acknowledgement. We address your remaining concerns below.
> > >
> > > ---
> > >
> > > ## 1) Regret analysis
> > >
> > > Building on the instantaneous regret bound from our rebuttal, language enters as a *margin term* $\kappa_t$, which has a clear physical meaning: it quantifies how much the language signal separates the optimal action from competing actions at round $t$.
> > >
> > > Under a fixed-objective surrogate, we now extend this to theoretically show that language introduces an additive margin $\kappa_t$, **improving cumulative regret by $-\sum_t \kappa_t$** while preserving the standard GP-UCB rate, and **reducing sample complexity by $O\left(\frac{1}{\epsilon} \sum_{t=1}^T \kappa_t\right)$**, as informative language eliminates suboptimal comparisons and accelerates learning, consistent with our empirical findings (Fig. 4, Table 1).
> > >
> > > **1. Cumulative regret.** Starting from the instantaneous bound introduced in the rebuttal above, summing over $t = 1, \dots, T$ gives
> > >
> > > $$R_T = \sum_{t=1}^T r_t \leq \sum_{t=1}^T \left[ 2\sqrt{\beta_t} \, \sigma_{t-1}(\cdot) - \kappa_t \right]\_+ + 2\sum_{t=1}^T \nu_t.$$
> > >
> > > Using standard GP-UCB analysis [1], and $\kappa_t^+ \leq 2\beta_t \sigma_{t-1}(\cdot) \forall t$, this yields
> > >
> > > $$R_T = \tilde{O}(\sqrt{T\gamma_T}) - \sum_{t=1}^T \kappa_t + 2\sum_{t=1}^T \nu_t$$
> > >
> > > Thus, informative language ($\kappa_t > 0$) reduces regret cumulatively, while misleading language ($\kappa_t < 0$) only affects it additively. $\kappa_t = 0$ recovers GP-UCB.
> > >
> > > **2. Sample complexity / convergence.** The cumulative improvement directly implies faster convergence. In standard GP-UCB [1],
> > >
> > > $$T_\text{UCB}(\epsilon) \sim \tilde{O}\left(\frac{\gamma_T}{\epsilon^2}\right).$$
> > >
> > > With language,
> > >
> > > $$T_\text{HEART}(\epsilon) \leq T_\text{UCB}(\epsilon) - O\left(\frac{1}{\epsilon} \sum_{t=1}^T \kappa_t\right),$$
> > >
> > > showing that language feedback reduces the number of required interaction rounds by accelerating posterior concentration.
> > >
> > > ---
> > >
> > > ## 2) Automated evaluation
> > >
> > > We designed a new experiment using Grok 4.1 Fast (1-shot) as the judge to evaluate latent preferences derived from a pre-collected human preference dataset - HelpSteer2 [2], thus eliminating structured metadata.
> > > ### 2.1) Validating the LLM-as-a-Judge
> > > To ensure this setup reliably mimics human evaluation, we measured agreement with human ground truth:
> > > | Match vs. Ground Truth | Helpfulness | Correctness | Coherence | Complexity | Verbosity |
> > > | :--- | :--- | :--- | :--- | :--- | :--- |
> > > | Within ± 1.0 | 100% | 80% | 100% | 100% | 100% |
> > > | Exact | 25% | 55% | 50% | 95% | 45% |
> > >
> > > This validates the LLM's reliability to serve as a proxy for human latent preferences.
> > > ### 2.2) Mimicking latent objectives
> > > We ran two experiments on 17 models, with 10 matching rounds and 5 evaluation rounds:
> > > - L1: Prefers complex answers; rating = complexity (95% exact accuracy).
> > > - L2: Prefers helpful, concise response; rating = helpfulness - verbosity (Spearman = 0.04, indicating near-independence).
> > >
> > > | Algorithm | L1 Cost | L1 Average Rating (0 to 4) | L2 Cost | L2 Average Rating (-4 to 4) |
> > > | :--- | :--- | :--- | :--- | :--- |
> > > | CUPID | 0.0399 ± 0.0028 | 3.6 | 0.0422 ± 0.0060 | 0.27 |
> > > | LMA | 0.0624 ± 0.0057 | 3.73 | 0.0684 | 0.2 |
> > >
> > > **Key findings:** CUPID reduces cost by 36% (L1) and 38% (L2) while achieving comparable or better quality (higher rating in L2, slightly lower in L1), supporting our thesis that it is especially effective when preferences are multi-dimensional and latent and aligning with our human study results.
> > >
> > > ---
> > >
> > > ## 3) Non-tech/High-stakes scenarios
> > >
> > > Agreeing with your concern, we provide new insights from previously unanalyzed data. Stratifying by self-reported AI familiarity, lower-familiarity participants (scores 2–3,  where 1 = rarely used, 5 = daily use) achieved CUPID quality ratings of 3.67–3.63, comparable to or exceeding LMA ratings of 3.20–4.00 from higher-familiarity users (scores 4–5). This suggests that CUPID's guided exploration compensates for limited user expertise, enabling non-expert users to achieve competitive performance.
> > >
> > > CUPID’s modular architecture (Figure 2) allows adding safeguards without modifying the core HEART-UCB algorithm: filtering fluent-but-inaccurate models prior to selection via a hard constraint in the budget controller; adding calibration accuracy $z_{\text{acc}}$ into the belief vector $b_t$; adding static expert-ranked weights $S=[s_1,\ldots,s_M]$ to the language-driven bias every round. We will add this discussion to the expanded limitations section.
> > >
> > > [1] Srinivas et al. Gaussian Process Optimization in the Bandit Setting: No Regret and Experimental Design. NeurIPS 2010.
> > >
> > > [2] NVIDIA. HelpSteer2: Open-source dataset for training top-performing reward models. https://huggingface.co/datasets/nvidia/HelpSteer2
> > >
> > > ---
> > >
> > > We would like to genuinely thank you again for your thorough review that helped us strengthen our paper. We hope that we have put all our efforts to address all your concerns in the best possible way for a more favorable score.

---

### Official Review · Reviewer_2ack · 2026-03-15

**Soundness:** 4
**Presentation:** 4
**Significance:** 2
**Originality:** 2
**Overall Recommendation:** 5
**Confidence:** 5

**Summary:**

This paper studies a user-to-LLM matching problem in which users have latent, hard-to-articulate objectives and limited cost/time budgets, and it proposes `CUPID`, a budget-aware active-learning framework based on dueling-bandit interactions. The core algorithm, `HEART-UCB`, combines GP-based preference uncertainty, a belief over latent objectives, optional language feedback, and cost penalties to choose informative model comparisons.

**Compliance With Llm Reviewing Policy:**

Affirmed.

**Final Justification:**

I will raise my score.

**Key Questions For Authors:**

- Could the authors evaluate `CUPID` on a more heterogeneous, cross-provider model zoo with output-level task signals, to show that the method generalizes beyond metadata matching within a single provider?
- Could the authors report how sensitive the language-feedback gains are to the specific auxiliary LLM used to generate arm-level indicator functions?

**Limitations:**

yes

**Strengths And Weaknesses:**

## Strengths
- The paper tackles a well-motivated problem that is distinct from standard per-query LLM routing: selecting one deployment model for a user/application under persistent latent preferences and explicit budget constraints. The paper articulates this distinction clearly and convincingly.
- The method is technically substantive. The paper formalizes latent-objective dueling bandits, defines a belief-aware UCB over objectives, and augments it with language-driven bias and cost penalties, which is a coherent and fairly novel combination.
- The paper goes beyond simulation by running blinded within-subject user studies for both text and image settings, and the results show a clear advantage for budget compliance with roughly comparable pooled quality, which is a meaningful empirical signal for a budget-aware matcher.

## Weaknesses
- In Section 4.1 and Appendix D, the main automated evaluation is conducted on a single-provider model zoo and uses mostly metadata-level, quantifiable objectives rather than response-level capabilities. This narrows the experimental problem relative to the paper's broader framing of latent user-to-LLM matchmaking on arbitrary model zoos [1,2,3].
- In Section 4.1 and Appendix D.2, the simulated user pipeline is highly structured: the judge has access to the hidden objective subset, and the user simulator emits exactly one directional preference at a time. This setup appears to be much simplified and cleaner than real user preference elicitation, where feedback is often inconsistent, multi-faceted, or absent [4,5].
- The paper presents posterior-prediction derivations and a belief-aware UCB construction, but I did not see a regret or sample-complexity guarantee for `HEART-UCB` under latent objectives, language bias, and cost penalties [1,3,6].



[1] Ong, Isaac, et al. "RouteLLM: Learning to Route LLMs from Preference Data." The Thirteenth International Conference on Learning Representations.

[2] Chiang, Wei-Lin, et al. "Chatbot arena: An open platform for evaluating llms by human preference." Forty-first International Conference on Machine Learning. 2024.

[3] Xia, Yu, et al. "Which llm to play? convergence-aware online model selection with time-increasing bandits." Proceedings of the ACM Web Conference 2024. 2024.

[4] Wang, Zhilin, et al. "HelpSteer2-Preference: Complementing Ratings with Preferences." The Thirteenth International Conference on Learning Representations.

[5] Zheng, Lianmin, et al. "Judging llm-as-a-judge with mt-bench and chatbot arena." Advances in neural information processing systems 36 (2023): 46595-46623.

[6] Bouneffouf, Djallel, and Raphael Feraud. "Multi-Armed Bandits Meet Large Language Models." AAAI Conference on Artificial Intelligence. 2026.

---

> ### Author Rebuttal · Authors · 2026-03-31
>
> We sincerely thank you for your review. Kindly see our response below.
>
> ---
>
> ## 1. Q1 & W1–W2: Cross-provider evaluation and simulation realism
>
> The controllability of the automated experiment is a deliberate design choice rather than a shortcoming, allowing us to isolate *algorithmic behavior* under well-defined (measurable) conditions. To further verify the robustness of simulation, we considered three types of language perturbations (Section 4.2). The broader claim is validated by a follow up human study. We would be very happy to run additional experimental setups if the reviewer has any suggestions.
>
> We clarify that we evaluate **37 models from 9 providers** (OpenAI, Anthropic, Google, Meta, Mistral, DeepSeek, xAI, MiniMax, MoonshotAI) in Section 4.2.
>
> **New experiment.** We additionally ran a new evaluation (same protocol as Section 4.1) on **17 models from 7 providers (xAI, Google, Anthropic, DeepSeek, Z.ai, Moonshot AI, MiniMax)** with MMLU-Pro scores **75.9–86.3**:
>
> | **Method** | **Cost (\$)** | **CR ↓** | **CP ↑** |
> |:---|:---|:---:|:---:|
> | **CUPID** | \$0.031 ± 0.005 | 3.2 | 100% |
> | LMA | \$0.045 ± 0.023 | 1 | 20% |
> | ε-greedy | \$0.026 ± 0.014 | 3.75 | 80% |
>
> CUPID achieves the highest convergence percentage (100%) with the lowest variance. Together with Figure 5 showing all CUPID variants outperforming baselines (+70.36% score, −86.48% cost vs. LMA), these results demonstrate cross-provider generalization.
>
> Regarding simulation realism: our aim for Section 4.1 is to confirm CUPID works with hard, tangible constraints before identifying latent objectives. We designed the experiment to receive one directional preference per round and continue neutral responses ("NONE") even after convergence—a harder setting for cost. The robustness experiment (Section 4.2) adds noisy/adversarial feedback, and the human study (Section 4.3) provides real-user validation with genuinely latent preferences, where CUPID outperforms in H3 and H4.
>
> ---
>
> ## 2. Q2: Auxiliary LLM sensitivity
>
> We reran experiment A1 and A2, observed the behavior and recorded the cost of the auxiliary model.
>
> *Table: Instructions Followed: the LLM consistently generated a string of binary tokens in the correct format. Constraints Followed: the generated string abides to the constraints of the direction text. Cost/Round: the average cost per round for this model. Accuracy: the accuracy of CUPID with the auxiliary LLM.*
>
> | **Attribute** | **Llama 3.3 70B** | **Qwen3 235B** | **Gemini 2.5 FL** | **Gemini 3 Flash Preview** | **o3** | **Grok 4.1 Fast (previously selected)** |
> |:---|:---:|:---:|:---:|:---:|:---:|:---:|
> | Instructions followed | ✗ | ✗ | ✗ | ✓ | ✓ | ✓ |
> | Constraints followed | ✗ | ✗ | ✗ | ✓ | ✓ | ✓ |
> | Cost/round | Open-source | Open-source | \$0.00033 | \$0.00201 | \$0.00871 | \$0.00191 |
> | CUPID accuracy | 100% | 100% | 100% | 100% | 100% | 100% |
>
> All models achieve 100% accuracy because when an auxiliary model fails to produce correctly formatted output, the system falls back to all-zeros (equivalent to "CUPID without language" from Section 4.1, which itself achieves 100% convergence, though with more iterations). While weaker models can be used, Grok 4.1 Fast consistently follows both the formatting instructions and constraint semantics while offering strong cost efficiency.
>
> Thank you so much for pointing this out, and we will add the discussion in our main text and appendix.
>
> ---
>
> ## 3. W3: Regret bounds
>
> Thanks! Yes, we have an elegant instantaneous regret bound that explicitly captures the effect of language through a margin term. Assume $L_t(a) \leq f(a) \leq U_t(a)$ and selection $a_t = \arg\max_a [U_t(a) + \phi_t(a)]$. Define $\kappa_t = \phi_t(a^\*) - \max_{a \neq a^\*} \phi_t(a)$. Then
>
> $$r_t \leq \left[ U_t(a_t) - L_t(a_t) - \kappa_t \right]_+.$$
>
> For GP-UCB,
>
> $$r_t \leq \left[ 2\sqrt{\beta_t}\sigma_{t-1}(a_t) - \kappa_t \right]_+.$$
>
> Thus, $\kappa_t > 0$ (informative) reduces regret, $\kappa_t = 0$ recovers standard UCB, and $\kappa_t < 0$ (misleading) increases regret only additively.
>
> ---
>
> **Literature.** We also appreciate additional literature. As discussed in our literature review, [1] is a query-level routing algorithm between only two LLMs. [2] is the main baseline through all our experiments. Interestingly, we started this line of research with [4]; however, it could not be used for automated experiments as its metrics are heavily correlated (0.94 for helpfulness-correctness, 0.5 for helpfulness-coherence, and 0.31 for complexity-verbosity).
>
> ---
>
> We again thank the reviewer for your positive comments and insights, and we hope that we addressed all of your concerns.

---

> > ### Author Rebuttal · Reviewer_2ack · 2026-04-04
> >
> > Thanks. I will raise my score.

---

> > > ### Author Response · Authors · 2026-04-07
> > >
> > > We thank the reviewer for their thoughtful engagement throughout the review process and for their vote of confidence. We appreciate their recognition of our work, and we will make the changes promised in our rebuttal in the camera-ready version.

---

### Official Review · Reviewer_zwEN · 2026-03-16

**Soundness:** 3
**Presentation:** 3
**Significance:** 2
**Originality:** 2
**Overall Recommendation:** 4
**Confidence:** 4

**Summary:**

The paper studies how to efficiently match users with suitable LLMs when their preferences are latent and difficult to specify. It proposes an interaction-efficient active learning framework that uses a dueling bandit algorithm to iteratively compare pairs of LLMs, collect user feedback, and update beliefs about user preferences through a belief-aware UCB strategy that balances exploration and exploitation under cost and time constraints. Experiments on multiple LLMs and human studies suggest that the approach can identify well-aligned models while reducing selection cost.

**Compliance With Llm Reviewing Policy:**

Affirmed.

**Final Justification:**

All my concerns have been addressed. Raised my score.

**Key Questions For Authors:**

1. The evaluation primarily uses 25 models from OpenAI, which may have relatively clear capability differentiation across releases. How would the proposed framework perform when the model pool contains highly competitive models from different providers (e.g., OpenAI, Anthropic, Google, Meta) whose capabilities and costs overlap more significantly? Have the authors considered evaluating CUPID in such a cross-provider setting?
2. The method relies on pairwise comparisons between models. How does the approach scale when the model zoo becomes much larger (e.g., hundreds of models)? Could the interaction cost or convergence time become prohibitive in such scenarios?
3. The algorithm incorporates language feedback using an auxiliary LLM to bias the UCB score. How sensitive is the system to incorrect or ambiguous interpretations by the auxiliary LLM? Have the authors evaluated cases where the language signal is misleading or inconsistent?

**Limitations:**

No specific limitations are discussed. The limitations could be better addressed if the concerns regarding model pool diversity and scalability to larger model pools are properly considered and evaluated.

**Strengths And Weaknesses:**

Strength:
1. Well-motivated problem setting: The paper addresses a practical challenge of selecting an appropriate LLM from a large model pool when user preferences are latent and difficult to articulate.
2. Structured algorithmic framework: The proposed CUPID framework combines dueling bandits, belief-aware UCB, and budget constraints to enable interactive preference learning for model selection.
3. Practical system implementation: The paper presents a UI-based tool that enables users to interactively compare model responses and provide feedback, demonstrating the feasibility of deploying the proposed framework in a real user-facing setting.

Weakness:
1. Limited novelty: Many components (e.g., dueling bandits, UCB exploration, GP-based preference modeling) build on existing techniques, and the contribution mainly lies in integrating them rather than introducing fundamentally new methods.
2. Limited analysis of scalability: The paper does not clearly discuss how the approach scales to very large model pools or more complex real-world deployment scenarios.
3. Evaluation limitations: The automated evaluation uses 25 models from OpenAI, whose capabilities and positioning across releases are often relatively distinct and may not strongly overlap. This setup may simplify the selection task. To better reflect realistic deployment scenarios, the evaluation should include models from multiple providers with closer capability overlap (e.g., comparisons across OpenAI, Anthropic, Google, etc., such as Claude Opus vs. GPT-series models).

---

> ### Author Rebuttal · Authors · 2026-03-31
>
> We sincerely thank you for your insightful reviews.
>
> ---
>
> ## Q1 & W3: Cross-provider evaluation
>
> We believe there may be a misunderstanding: the robustness experiment (Section 4.2) already evaluates **37 models from 9 providers** (including OpenAI, Anthropic, Google, Meta, Mistral, DeepSeek, xAI, MiniMax, MoonshotAI) with diverse capabilities through the MMLU Pro and the AIME 2025 benchmarks ($R^2 = 0.52$).
>
> **New experiment.** Additionally, we have now conducted new experiments (same protocol as in Section 4.1) to evaluate **17 models from 7 providers (xAI, Google, Anthropic, DeepSeek, Z.ai, Moonshot AI, and MiniMax)** with MMLU Pro scores from **75.9-86.3**, representing a "model pool contains highly competitive models from different providers whose capabilities and costs overlap more significantly." We excluded vanilla latent dueling bandits and RUCB because they were evidently much weaker than the standards.
>
> | **Method** | **Cost (\$)** | **CR ↓** | **CP ↑** |
> |:---|:---|:---:|:---:|
> | **CUPID** | \$0.031 ± 0.005 | 3.2 | 100% |
> | LMA | \$0.045 ± 0.023 | 1 | 20% |
> | ε-greedy | \$0.026 ± 0.014 | 3.75 | 80% |
>
> The heightened cost can be accounted for by the routing model, which takes roughly 30% of the total cost (see Part 1 of our response to Reviewer n6nD). Across all baselines, it can be seen that CUPID gives the most consistent result, lowest uncertainty and highest CP.
>
> ---
>
> ## Q2 & W2: Scalability to large model pools
>
> With the expansion of the model pool, current practice increasingly relies on know-how, which motivates the need for a principled method such as ours. However, we agree that the GP preference model has complexity $O(M^3)$, which is not a bottleneck for a pool with $M < 1000$ models. Scalability can be further improved by,
> 1. Algorithmically: a Sparse GP solution [1].
> 2. Prior knowledge: limiting the number of models fed into the LLM usage based on past comparisons (e.g. excluding models that constantly lost duels).
>
> We will add this discussion to the revision.
>
> ---
>
> ## Q3: Sensitivity to incorrect language feedback
>
> We have provided results on the sensitivity towards 1) incorrect interpretations, 2) misleading language, or 3) inconsistent language in Section 4.2. Figure 5 shows that all three CUPID scenarios outperforms the two baselines, with the strongest results being **+70.36% in score** and **-86.48% in cost** compared to LMA.
>
> We also reran experiment A1-A2 with five additional auxiliary models (kindly see our response to Reviewer 2ack).
>
> ---
>
> ## W1: Novelty of approach
>
> We respectfully note that identifying entirely novel algorithmic primitives in the classical sense has become increasingly rare at NeurIPS and ICML. Since the contribution landscape has evolved toward principled integration of ideas and new problem formulations, we believe our work as a coherent and carefully designed synthesis to solve an increasingly challenging problem. Even in this context, we propose (i) the first formulation of **latent dueling bandits with UCB**, where the objective itself is unknown and must be inferred online and (ii) a mechanism for **incorporating natural language feedback into UCB** as a bounded, adaptive bias. As each component interacts with each other (e.g., the language bias must respect the belief-aware UCB structure, and the cost penalty must not collapse exploration), we believe the novelty lies in the formulation and coherent integration, not in any single component. The tight coupling, where language bias must remain consistent with belief-aware UCB and cost penalties must preserve exploration, makes the overall formulation a fundamentally challenging problem. In Section 4.1, HEART-UCB's own components used as standalone baselines (latent dueling bandits, ε-greedy) show inconsistent convergence, while even CUPID's ablations converge in all runs—demonstrating the integrated system is substantially more than the sum of its parts.
>
> [1] Titsias, M. (2009). Variational Learning of Inducing Variables in Sparse Gaussian Processes. *AISTATS'09*.
>
> ---
>
> We again thank you for your insights, and we hope we have addressed all of your concerns.

---

> > ### Author Rebuttal · Reviewer_zwEN · 2026-04-02
> >
> > While the rebuttal improves the evaluation by adding cross-provider experiments, it does not sufficiently address concerns on scalability and novelty; in particular, there is no measurement of actual runtime or time-to-selection, leaving the practical time overhead unexamined, so I maintain my original score. I would raise my score if the authors can share some results regarding the practical time overhead.

---

> > > ### Author Response · Authors · 2026-04-07
> > >
> > > Thank you for specifying the remaining concern. We address your remaining concerns below:
> > >
> > > ---
> > >
> > > ## Scalability and practical time overhead
> > > CUPID makes exactly 3 LLM API calls per round (1 auxiliary + 2 candidates), independent of pool size $M$. The GP update is $O(M^3)$, completing in <1s for $M \leq 1000$. Total time-to-selection is $T \times$ (auxiliary latency + max candidate latency), dominated entirely by LLM response times, not CUPID's algorithm.
> > >
> > > We now ran four experiments scaling from $M = 5$ to $M = 125$ models across up to 18 providers, using HelpSteer2 prompts. We used Gemini 3 Flash Preview, having similar performance as Grok 4.1 Fast, as the auxiliary model for its low latency at the time we conducted this experiment (see our response to Reviewer 2ack).
> > >
> > > - **Experiment V1 (M = 125):** We expanded the OpenAI model zoo to 125 models from 18 providers (OpenAI, Google, Anthropic, Meta Llama, Microsoft, Amazon, xAI, Perplexity, DeepSeek, Qwen, Mistral AI, NVIDIA, MiniMax, Moonshot AI, Z.ai, Xiaomi, StepFun, Inception) and shuffled it to verify runtime scalability.
> > >
> > > - **Experiment V2 (M = 75):** The first 75 models from V1.
> > >
> > > - **Experiment V3 (M = 25):** The OpenAI model zoo.
> > >
> > > - **Experiment V4 (M = 5):** The last 5 models from V1.
> > >
> > > ---
> > >
> > > ### Scalability with the number of models
> > >
> > > | Practical time in seconds | M = 125 | M = 75 | M = 25 | M = 5 |
> > > |---|---:|---:|---:|---:|
> > > | CUPID overhead (without aux. LLM) | 0.719 | 0.267 | 0.094 | 0.035 |
> > > | CUPID overhead (with aux. LLM) | 4.618 | 2.595 | 3.485 | 1.833 |
> > > | Time-to-selection (without aux. LLM) | 8.532 | 6.196 | 15.200 | 16.414 |
> > > | Time-to-selection (with aux. LLM) | 12.431 | 8.524 | 18.591 | 18.212 |
> > >
> > > ---
> > >
> > > ### Detailed breakdown of each component
> > >
> > > | Stage | V1 (M = 125) | V2 (M = 75) | V3 (M = 25) | V4 (M = 5) |
> > > |---|---:|---:|---:|---:|
> > > | GP update | 0.172 | 0.084 | 0.052 | 0.023 |
> > > | UCB scoring | 0.547 | 0.183 | 0.042 | 0.012 |
> > > | Belief update | <0.0001 | <0.0001 | <0.0001 | <0.0001 |
> > > | LLM duel calls | 7.813 | 5.929 | 15.106 | 16.379 |
> > > | Auxiliary LLM (Gemini 3 Flash Preview) | 3.899 | 2.328 | 3.391 | 1.798 |
> > >
> > > ---
> > >
> > > **Key observations:** (1) CUPID's algorithmic overhead (GP + UCB + belief) scales with $M$ but remains **<1 second even at M = 125**—negligible compared to LLM generation time. (2) Time-to-selection is dominated by LLM API latency and actually *decreases* at larger $M$ in our experiments (V1/V2 are faster than V3/V4), because the model pool happened to contain faster-responding models. (3) The auxiliary LLM adds 2 - 4 seconds per round regardless of $M$, confirming it does not introduce a scalability bottleneck.
> > >
> > > Thank you for your feedback, and we will include wall-clock measurements in the revision.
> > >
> > > ---
> > >
> > > ## Novelty
> > > We additionally would like to provide some contexts to the recent bandit literature. Recent contributions at top venues consistently follow patterns of (i) integration of techniques [1, 2], (ii) principled application to new domains  [3, 4, 5], or (iii) additive extension of classical methods [6, 7], to which, in our view, we contribute through all three: (i) formulating latent dueling bandits with UCB, (ii) principled integration into CUPID for user-level LLM matching, (iii) the bias, penalty, and cooldown mechanisms as additive extensions as suited for the application at hand. We remain open to specific feedback on what further novelty clarification would be helpful.
> > >
> > > [1] Di et al. Nearly Optimal Algorithms for Contextual Dueling Bandits from Adversarial Feedback. ICML 2025.
> > >
> > > [2] Wang et al. Online Clustering of Dueling Bandits. ICML 2025.
> > >
> > > [3] Li et al. A Contextual-Bandit Approach to Personalized News Article Recommendation. WWW 2010.
> > >
> > > [4] Agarwal et al. Multi-Armed Bandits with Network Interference. NeurIPS 2024.
> > >
> > > [5] Kong et al. Improved Analysis for Bandit Learning in Matching Markets. NeurIPS 2024.
> > >
> > > [6] Badanidiyuru et al. Bandits with Knapsacks. JACM 2018.
> > >
> > > [7] Zhang & Cheung. Piecewise-Stationary Bandits with Knapsacks. NeurIPS 2024.
> > >
> > > ---
> > >
> > > We again thank the reviewer for your insights. We hope that we have put all our efforts to address all your concerns in the best possible way for a more favorable score.

---

### Decision · Program_Chairs · 2026-04-30

**Decision:**

Accept (regular)

**Comment:**

After extensive rebuttal all reviewers are now positive on this paper.  the recommendations are 3 Weak Accepts and one Accept.

Reasons to accept the paper:  This paper addresses a practical challenge of matching a particular LLM model to a user's preference when the user's preference is latent and difficult to articulate.  They introduce a structured algorithmic framework, called CUPID that combines dueling bandits, belief aware UCB and budget constraints. The paper validates the cost savings of the algorithm with a real user preference study.

Reasons to reject this paper include primarily the limited novelty.  Many components (e.g., dueling bandits, UCB exploration, GP-based preference modeling) build on existing techniques, and the contribution mainly lies in integrating them rather than introducing fundamentally new methods.  While the authors rebut that theirs is the "first formulation of latent dueling bandits with UCB, where the objective itself is unknown and must be inferred online " and a includes mechanism for incorporating natural language feedback into UCB.

Since the model does A/B testing it can only evaluate models in a pairwise fashion so the scalability of this method across a large number of models was questioned.  Even after rebuttal one reviewer maintains that the system does not sufficiently address concerns on scalability and novelty; in particular, there is no measurement of actual runtime or time-to-selection, leaving the practical time overhead unexamined.

The user preference study participant pool consists of a convenience sample of highly educated young people with the majority studying computer science (limited diversity of users) and the participants are given a preference rubric on which to evaluate the LLMs which may not include all the true "latent variables" that determine user preferences.

Limited analysis of scalability: The paper does not clearly discuss how the approach scales to very large model pools or more complex real-world deployment scenarios.
Evaluation limitations: The automated evaluation uses 25 models from OpenAI, whose capabilities and positioning across releases are often relatively distinct and may not strongly overlap. This setup may simplify the selection task. To better reflect realistic deployment scenarios, the evaluation should include models from multiple providers with closer capability overlap (e.g., comparisons across OpenAI, Anthropic, Google, etc., such as Claude Opus vs. GPT-series models).